# The Centrosome and the Primary Cilium: The Yin and Yang of a Hybrid Organelle

**DOI:** 10.3390/cells8070701

**Published:** 2019-07-10

**Authors:** Vladimir Joukov, Arcangela De Nicolo

**Affiliations:** 1N.N. Petrov National Medical Research Center of Oncology, 197758 Saint-Petersburg, Russia; 2Veneto Institute of Oncology IOV-IRCCS, 35128 Padua, Italy

**Keywords:** centrosome, centriole, primary cilia, mitosis, cell cycle, microtubule cytoskeleton, microtubule nucleation, cell differentiation, organelle biogenesis, cell signaling

## Abstract

Centrosomes and primary cilia are usually considered as distinct organelles, although both are assembled with the same evolutionary conserved, microtubule-based templates, the centrioles. Centrosomes serve as major microtubule- and actin cytoskeleton-organizing centers and are involved in a variety of intracellular processes, whereas primary cilia receive and transduce environmental signals to elicit cellular and organismal responses. Understanding the functional relationship between centrosomes and primary cilia is important because defects in both structures have been implicated in various diseases, including cancer. Here, we discuss evidence that the animal centrosome evolved, with the transition to complex multicellularity, as a hybrid organelle comprised of the two distinct, but intertwined, structural-functional modules: the centriole/primary cilium module and the pericentriolar material/centrosome module. The evolution of the former module may have been caused by the expanding cellular diversification and intercommunication, whereas that of the latter module may have been driven by the increasing complexity of mitosis and the requirement for maintaining cell polarity, individuation, and adhesion. Through its unique ability to serve both as a plasma membrane-associated primary cilium organizer and a juxtanuclear microtubule-organizing center, the animal centrosome has become an ideal integrator of extracellular and intracellular signals with the cytoskeleton and a switch between the non-cell autonomous and the cell-autonomous signaling modes. In light of this hypothesis, we discuss centrosome dynamics during cell proliferation, migration, and differentiation and propose a model of centrosome-driven microtubule assembly in mitotic and interphase cells. In addition, we outline the evolutionary benefits of the animal centrosome and highlight the hierarchy and modularity of the centrosome biogenesis networks.

## 1. Introduction: On the Definition of the Centrosome

The architecture and motility of eukaryotic cells are underpinned by the dynamic and interconnected networks of actin and microtubule (MT) cytoskeletons [1,2]. MTs are polar, hollow filaments assembled from α/β-tubulin heterodimers in a GTP-dependent manner. The polymerization of α/β-tubulin subunits occurs in a head-to-tail fashion, resulting in the formation of 13 laterally associated protofilaments that make up the MT wall. In cells, MT ends with exposed β-tubulin (plus ends) are the preferential sites of MT elongation through the addition of α/β-tubulin, whereas MT ends with exposed α-tubulin (minus ends) are often stabilized and anchored at MT-organizing centers (MTOCs) [3,4,5].

The main MTOC is the centrosome, which is commonly defined as a non-membrane-bound organelle consisting of a pair of centrioles and the surrounding pericentriolar material (PCM) [6,7,8,9]. Centrosomes are involved in a variety of cellular processes, including MT and actin cytoskeleton organization, spindle assembly, intracellular signaling and trafficking, the establishment of cell polarity, cell motility, protein homeostasis, and immune response [2,6,10,11,12,13,14,15]. As a reflection of the important role of centrosomes in cell physiology, mutations in numerous centrosomal proteins cause various disorders, including microcephaly, dwarfism, ciliopathies, and other pathologies associated with degeneration of neural and other tissues [11,16,17,18,19]. Moreover, structural, functional, and/or numerical centrosomal abnormalities are a hallmark of most cancers [11,17,20,21,22].

Over a century ago, Theodor Boveri, the “father” of centrosome research, described the centrosome—as inferred in its name—as the dynamic center and the “division organ” of the cell, noting that the division of the centrosome creates the centers of the forming daughter cells, around which other components are organized symmetrically [23]. Boveri also observed that the animal centrosome consists of two main components: the internally located granular centriole, which duplicates before the centrosome, and the outer centroplasm (now known as PCM), which organizes and anchors “astral rays” [23]. It was subsequently established that the “astral rays” represent MTs anchored at the PCM, and the term MTOC was introduced by Pickett-Heaps to designate structures from which MTs emanate [24,25]. With the discovery of γ-tubulin, a conserved subtype of tubulin and a key MT-nucleating component of animal centrosomes and the acentriolar yeast spindle pole bodies [26,27,28,29,30,31], the term MTOC was often used instead of, or even synonymously to centrosome. It was shown, however, that, besides centrosomes, a number of other cellular structures also promote γ-tubulin-mediated MT nucleation and anchoring [32,33,34,35]. Moreover, in many unicellular eukaryotes, the only prominent site of MT anchoring is represented by the basal bodies, structures analogous to centrioles, which associate with the plasma membrane and form cilia or flagella–antenna-like extensions involved in locomotion, feeding, and sensation [36,37]. Hence, the basal body complex/apparatus of protists is sometimes considered as a centrosomal MTOC [37,38,39]. Given these semantic ambiguities, it is important to clarify the terms centrosome, MTOC, centriole, and basal body.

Boveri remarkably accurately captured essential characteristics of the centrosome, which form the basis for the definition of this organelle. In modern terms, the centrosome can be defined as an organelle with three basic properties: i) ability to form an MTOC at the center of the cell through γ-tubulin-dependent nucleation and anchoring of MTs at their minus ends; ii) ability to associate with the nucleus in interphase and spindle poles during mitosis; iii) duplication once during the cell cycle [40,41,42]. As described below, the properties (i) and (ii) of the animal centrosome (i.e., the localization to the center of the cell and the association with the nucleus) may not manifest in some instances, e.g., during primary ciliogenesis or immune synapse formation. Under the aforementioned definition, the spindle pole bodies of yeasts and the nucleus-associated bodies of amoebas, which lack centrioles, should be considered as centrosomes, whereas the basal body complex of most protists should not because it does not localize to the cell center (although in some organisms, the basal bodies are connected to the nucleus by specialized fibers and/or associate with spindle poles during mitosis–see below) (Figure 1 and Figure 2). The centriole/basal body is an evolutionary conserved cylindrical structure composed of nine symmetrically arranged triplets (or, in some organisms, doublets or singlets) of stable MTs [43,44]. As justified below (Section 4 and Section 5), when this structure is coated with the PCM and is involved in the formation of a centrosome or a primary cilium, the term centriole is used henceforth. Accordingly, when the structure lacks the PCM (and its main marker and key factor of centrosome biogenesis in animals, the centrosomal protein (CEP) of 192 kDa (CEP192) [45,46,47,48,49,50]) and nucleates a motile cilium/flagellum, the term basal body is used.

Lüders and Stearns defined MTOCs as structures that can catalyze γ-tubulin-dependent MT nucleation and can anchor MTs through their minus ends, plus ends, or sides [32]. This definition does not include structures, which may organize or nucleate MTs in a γ-tubulin-independent manner [51,52,53]. Among such structures are kinetochores, macromolecular assemblages formed at the centromeric chromatin, which mediate chromosome attachment to spindle MTs and regulate chromosome segregation [54]. Kinetochores participate in spindle assembly through a pathway that is distinct from that used by non-centromeric chromatin: they promote MT stabilization and bundling and anchor MTs through their plus ends [55,56,57,58,59,60,61,62,63]. Therefore, kinetochores could, in principle, be considered as MTOCs. Hence, we suggest a definition of MTOC as any structure that generates, organizes, and/or anchors MTs.

## 2. Evolutionary Origin of Centrosomes

As revealed by comparative genomic and phylogenetic analysis, canonical centrosomes—that is, those consisting of one or two centrioles and the surrounding PCM—are found in the Amorphea (Unikonts) supergroup, including all animal lineages and certain lower fungi (chytrids) and amoebas. In addition, canonical centrosomes are found in some early-branching eukaryotes of the SAR (stramenopiles, alveolates, and Rhizaria) supergroup (Figure 1). Unlike centrosomes, centrioles are widespread across eukaryotes in the form of basal bodies that organize motile cilia/flagella [40,45,64,65].

This fact suggests that centrioles evolved independently of centrosomes and were secondarily incorporated into the latter [40,42,67]. Tracing back the evolution of centrosomes may provide clues to understanding the biogenesis, role, and functions of these organelles.

Cavalier-Smith proposed that the pre-eukaryotic ancestor had a precursor centrosome, which associated both with the plasma membrane and with chromatin and had a dual centrosome/kinetochore function in segregating the duplicated DNA [42] (Figure 1A). The subsequent phagotrophy-induced internalization of the DNA-membrane attachments may have imposed selective pressure for the evolution of mitosis wherein individualized chromosomes are segregated by a bipolar spindle. With the emergence of the proto-nuclear envelope, the precursor centrosome may have duplicated into two distinct MTOCs: one for cell-surface cortical MTs and the other one for the nuclear membrane-associated spindle poles to segregate chromosomes during closed mitosis. The two MTOCs were connected to one another and to the nucleus by centrin-containing fibers [42,67,68]. The plasma membrane-associated MTOC may have further evolved into a flagellar/ciliary apparatus, with one or two basal bodies nucleating a flagellar/ciliary axoneme(s) and an array of cortical MTs termed MT roots or flagellar roots [36,42,69]. It was inferred that the last common ancestor of all eukaryotes had a basal body apparatus similar to that of the modern-day eukaryotes of the Excavata supergroup for the formation of the motile cilium/flagellum and cortical MTs, and a second, nucleus-associated MTOC for bipolar spindle assembly. The two MTOCs were connected by centrin-containing fibers. An analogous MT cytoskeleton architecture may have been preserved in the last common ancestor of all Amorphea [36,40,42,67,69] (Figure 2).

The physical connection between the basal body apparatus and the nucleus-associated MTOC is found in several eukaryotic lineages in the form of the nucleus-basal body connector, or rhizoplast [72,73,74,75,76,77] (Figure 1B,D and Figure 2). This connection enables coordination and synchronization of the division of both MTOCs with that of the nucleus and the cytoplasm, which is essential for successful cell division. In addition, the connection may serve as a means of communication between the extracellular and intracellular domains. During semi-open mitosis in green algae *Chlamydomonas reinhardtii* and in *Giardia lamblia,* that belong to the Archaeplastida and Excavata supergroups, respectively, the flagella and the cytoplasmic MTs disassemble, and the basal bodies internalize and move towards spindle poles [37,73,78,79,80] (Figure 1D). In *G. lamblia,* the basal bodies merge with spindle poles, effectively forming MTOCs analogous to canonical mitotic centrosomes [73,80]. In green algae, the contact between the basal bodies and the nucleus is facilitated by the pre-mitotic contraction of the nucleus-basal body connector [37,75,81]. In the fresh-water golden alga *Ochromonas danica* (SAR supergroup) and in *Trichomonas vaginalis* (Excavata supergroup), which have open mitosis and closed extranuclear mitosis, respectively, the nucleus-basal body connector itself serves as a spindle pole-organizing MTOC [39,82] (Figure 1B). In brown algae (SAR supergroup), motile gametes have a basal body apparatus, which nucleates a pair of flagella and is connected to the nucleus by centrin-containing fibers. By contrast, brown algal vegetative cells lack flagella and cortical cytoskeleton, and, instead, have canonical centrosomes similar to those of animals in their overall appearance and behavior (Figure 1C) [83].

Thus, many extant eukaryotes have two types of primary MTOCs with distinct roles during alternate stages of the cell cycle: i) the basal body apparatus, which nucleates a motile cilium/flagellum and rootlet MTs, when cells are in a quiescent state; ii) one or two nucleus-associated MTOCs, which are often dormant in interphase, but form in mitosis and play an essential role in bipolar spindle formation. Furthermore, in some eukaryotes, during mitosis, the flagella and cytoplasmic MTs disassemble, and the basal bodies internalize and associate with the nucleus-associated MTOCs at spindle poles. This process may be driven by contraction of the nucleus-basal body connector and may effectively result in a transient formation of a canonical centrosome. 

## 3. The Animal Centrosome as a Symbiotic Composite of Two Distinct Functional Modules

On the basis of phylogenetic evidence, it was proposed that the animal centrosome evolved by direct filiation from the ancestral basal body complex, through its internalization and acquisition of the ability to recruit the PCM [40,68]. However, it seems more likely that the animal centrosome evolved through internalization of the ancestral plasma membrane-associated basal body complex and its merger with the ancestral juxtanuclear MTOC involved in spindle pole assembly. These two MTOCs may have been the precursors of the centrioles and the PCM, respectively. The main argument supporting this hypothesis is that the spindle pole-organizing MTOCs likely evolved before the cilia/flagella (Figure 1A) because of the basic, essential role of spindle poles in segregating chromosomes, organelles, and other cellular contents [42]. Indeed, whereas certain eukaryotic lineages are devoid of cilia/flagella, the spindle pole-organizing MTOCs are found in all eukaryotes, including those lacking centrosomes or conspicuous interphase MTOCs. In some protists of the SAR supergroup (e.g., *Plasmodium falciparum* and *Tetrahymena thermophila*) and the Excavata supergroup (e.g., *Trypanosoma brucei*), which lack discernible centrosomes and use basal bodies only for the assembly of cilia/flagella, the acentriolar mitotic spindle poles are organized inside the nucleus by specific nuclear membrane-associated formations: the centriolar plaques (*P. falciparum*), the laminar arrangements resembling the yeast spindle pole bodies (*T. thermophila*), and the ring-like structures adjacent to the inner side of the nuclear envelope (*T. brucei*) [84,85,86,87]. Even in higher plants, which lack basal bodies and centrosomes (presumably, due to a secondary loss) (Figure 1E), prior to nuclear envelope breakdown, MTs are assembled into distinct cytoplasmic MTOCs (polar caps, gametosomes, polar organizers, or the axial MT system), which play a role similar to that of centrosomes in organizing spindle poles [88]. Two other observations also support the notion that the centrioles and the PCM derived from two distinct ancestral MTOCs. First, the evolution of the nucleus-associated MTOC and the basal body apparatus was likely driven by completely different selective forces: to organize the mitotic spindle and to assemble a motile cilium/flagellum for cell locomotion, feeding, and sensation, respectively. This fact makes it unlikely that the MT-nucleating PCM originated from the basal body apparatus. Second, the animal centrosomes and the acentriolar centrosomes of yeasts and amoebozoans share many conserved proteins involved in PCM organization and in the anchoring of γ-tubulin complexes [67,89,90,91,92], supporting the origin of the PCM from the ancestral nucleus-associated MTOC.

Thus, the animal centrosome appears to be a composite organelle evolved from a merger between the ancestral nuclear membrane-associated MTOC and the plasma membrane-bound basal body apparatus, which were the precursors of the PCM and centrioles, respectively. The merger may have initially been transient and driven by the centrin-containing fibers connecting the two ancestral MTOCs. As the merger became permanent in mitosis and interphase, the fibrous connector may have been lost (or replaced with a new type of the centrosome connector: see below—Section 8), and the centrosome may have acquired the ability to relocate and act as an MTOC at both ends of the connector, i.e, at the nuclear membrane and the plasma membrane (Figure 1G and Figure 2).

## 4. Building the Hybrid Organelle: The Centrosome Cycle

### 4.1. Overview of the Centrosome Cycle

The centrioles and the PCM have evolved an intimate symbiotic relationship and mutual dependence on each other so that in most proliferating cells, the two structures can exist only as a composite organelle–what M. Bornens has recently referred to as “the primary cilium/centrosome organ” [68,93,94,95]. Moreover, mechanisms have evolved that endow the PCM with the ability to assemble a single centriole per centrosome during each cell cycle and enable subsequent coating of the newly-formed centrioles with the PCM [6,94,96,97,98,99,100]. This symbiotic relationship between the centrioles and the PCM is evidenced by the centrosome cycle, a process during which centrosomes duplicate while performing their functions specific to each cell cycle phase.

Centrosome duplication is usually synchronized with the DNA replication cycle, and both cycles are driven by oscillations of cyclin and cyclin-dependent kinase (CDK) activity [6,100]. A typical animal cell begins the cell cycle with two centrosomes, each containing one centriole (Figure 3). The centrioles are connected through their proximal ends (a phenomenon called centrosome cohesion) with a flexible linker formed by rootletin, its paralog C-NAP1 [centrosomal never in mitosis A (NIMA)-related kinase 2 (NEK2)-associated protein 1, also known as CEP250], and CEP68 [101,102,103,104,105]. Additional proteins, such as leucine-rich repeat-containing protein 45 (LRRC45), centlein, and coiled-coil domain-containing protein 102B (CCDC102B), have been implicated in linker formation [106,107,108]. The centriole of the older (mother) centrosome contains, at its distal end, subdistal and distal appendages. The distal appendages promote membrane docking and are essential for the formation of the primary cilium, a specialized solitary non-motile cilium found in most animal cells, which detects and transmits extracellular cues to regulate diverse cellular functions (Figure 4). The subdistal appendages are thought to aid in the positioning of the primary cilium through anchoring cytoplasmic MTs [10,100]. In interphase, the mother and daughter centrosomes often coalesce into one juxtanuclear MTOC, although early in the cell cycle, the daughter centrosome may migrate throughout the cytoplasm, while remaining leashed to the mother centrosome by the linker. Such behavior of the daughter centrosome was attributed to a transient loss of its ability to anchor MTs (with retained MT-nucleating activity) [109].

Centrosome duplication can be subdivided into two main, temporally overlapping stages – centriole assembly and centriole-to-centrosome conversion (CCC) (Figure 3). The centriole assembly is initiated in G1-early S phase and continues through G2 phase. During this process, a new, orthogonally oriented centriole, termed a procentriole, is formed at the proximal end of each of the two parental centrosomes [43,100]. The CCC is a multi-stage process, during which the procentriole acquires the PCM and becomes a fully functional centrosome capable of serving as an MTOC and of duplicating [95,97,99,110,111,112,113]. The CCC begins in S phase, continues into G2 and M phases, and completes in the G1 phase of the next cell cycle. As such, the CCC encompasses all events pertaining to the ancestral, mitotic role of centrosomes in spindle assembly and chromosome segregation.

In G2 phase, centrosomes dramatically increase their size and MT-nucleating capacity through the recruitment of additional PCM components. Morphologically, this process, termed centrosome maturation, comprises the formation of the outer, mitotic PCM layer, over the interphase PCM layer [114,115] (Figure 3).

The mitotic PCM enables nucleation and anchoring of MTs and their organization in a radial array, known as the MT aster. Centrosome maturation is a prerequisite for the concomitantly occurring process of centrosome separation, and both processes are essential for bipolar spindle assembly. During centrosome separation, the centrosome cohesion is dissolved through the breakage of the intercentrosomal linker, and the two centrosomes separate to form spindle poles [116] (Figure 3). The centrosome linker dissolution is mediated by the NEK2A kinase, which is activated by Polo-like kinase 1 (PLK1; Polo in *D. melanogaster*)—the founding member of the Polo-like kinase (PLK) family—and phosphorylates linker components, promoting their removal [101,102,107,108,117,118]. The centrosomes are then moved apart by the MT-sliding activity of the plus end-directed motor proteins kinesin family member 11 (KIF11) (also known as kinesin-5 and Eg5) and 15 (KIF15) [119,120,121,122]. 

As cells exit mitosis, the mitotic PCM layer disintegrates, and each daughter centriole disengages from the parental centrosome while retaining a connection to it with a newly formed flexible linker [98,99,123,124,125,126,127]. The passage through mitosis also ensures that the parental centriole in both daughter cells acquires/restores the subdistal and distal appendages (which deteriorate during mitosis) [128,129]. Thus, each nascent daughter cell inherits a pair of centrosomes—each containing a single centriole—the parental one, and the one formed by the daughter centriole that has completed the CCC (Figure 3). In interphase animal cells, the two centrosomes appear as a single MTOC (in fact, often referred to as a single centrosome) because the daughter centrosome may transiently lose its ability to anchor MTs and because the centrosomes are closely juxtaposed by a linker and do not separate until G2 phase [109,116,130,131]. For comparison, in brown algae, two oppositely placed centrosomes are visible during most of the interphase because centrosome duplication and separation occur soon after cytokinesis [83] (Figure 1C, left panel, compare to Figure 1G, middle panel). By contrast, yeast cells have a single acentriolar centrosome in G1 phase [91] (Figure 1F, left panel). The unequal ability of the two centrosomes in animals to nucleate MTs and a primary cilium is the result of generational asymmetry, which plays an important role in centrosome biogenesis and function [10,132,133].

The fact that the two stages of centrosome duplication comprise the generation of a new centriole and its coating with the PCM implies that the centrosome cycle has evolved through the integration of two distinct molecular modules, namely, the core basal body assembly module and the PCM assembly module, respectively, with the cell cycle machinery. The basal body assembly module is conserved across eukaryotes, whereas the PCM assembly module is specific to Amorphea and appears to have undergone a substantial evolution within this clade [6,9,40,43,44,45,46,65,67,115]. Among centrosomal proteins, the presence of the scaffold protein CEP192 [spindle-defective protein 2 (Spd-2) in invertebrates] is most strongly correlated with the presence of canonical centrosomes in the organism, implying that this protein has played a key role in the evolution of these organelles [40,45,46]. Consistent with this notion, CEP192 is the key regulator of PCM formation and is essential for both stages of centrosome duplication, as well as for centrosome maturation [48,49,134,135,136,137,138,139,140,141].

### 4.2. Centriole Assembly

Centriole assembly is initiated in G1-early S phase through localized recruitment and concentration of the Polo-like kinase 4 (PLK4) (Sak in *Drosophila melanogaster*; ZYG-1 in *Caenorhabditis elegans*) in a confined area of the PCM in the proximal end of each of the two parental centrosomes [100,142,143,144]. The recruitment is mediated by CEP192 and two other centrosomal scaffold proteins, CEP152 [asterless (Asl) in *D. melanogaster*] and CEP63. CEP192/Spd-2 is essential for the centrosomal accumulation of PLK4 in humans and *C. elegans*, but not in *D. melanogaster*, in which CEP152/Asl recruits PLK4 through direct interaction [135,136,139,140,145,146,147,148,149,150,151]. It was shown that in human cells, CEP192 and CEP152 cooperate in promoting PLK4 centrosome recruitment through direct binding to the kinase [135,136]. It should be noted in this regard that CEP192 promotes centrosome duplication also because it enables PCM formation, which is a prerequisite for centriole assembly and maintenance, as well as for the CCC [48,49,97,112,123,124,134,139,140,152]. The local PLK4 concentration in the PCM of the parental centrosomes promotes PLK4 activation through trans-autophosphorylation [96,153,154], which initiates the sequential recruitment of a set of highly conserved centriolar proteins (Figure 5, left panel). PLK4 is activated in a concentration-dependent, autocatalytic manner [96,155,156]. The phosphorylation by PLK4 of its binding partner, SCL-interrupting locus protein (STIL) [anastral spindle 2 (Ana-2) in *D. melanogaster*; spindle assembly abnormal protein 5 (SAS-5) in *C. elegans*] promotes the recruitment of SAS-6, an essential component of the cartwheel–a 9-fold symmetrical structure, which establishes and stabilizes MT triplets of the centriole wall ([43,44,100,157,158] and references therein). The centrosomal P4.1-associated protein (CPAP) [also known as centromere protein J (CENPJ); SAS-4 in *D. melanogaster* and *C. elegans*] and CEP135 (also known as BLD10) then promote centriole elongation by aiding the polymerization of centriolar MTs in a process involving several members of the tubulin family (Figure 5, left panel). Centriole elongation continues through S and G2 phases (reviewed in [43,44,100,158]).

Recent studies imply that centriole assembly sets the stage for the CCC through the centrosomal protein CEP295 (SAS-7 in *C. elegans;* Ana1 in *D. melanogaster*) [95,110,112,113]. CEP295 localizes to the inner layer of the proximal part of the parental centrosomes, from where it is recruited to the wall of the newly-formed procentrioles [112]. CEP295 then recruits CEP192, presumably through direct binding [95,112,113] (Figure 5, left panel).

A novel centrosomal protein PPP1R35 (protein phosphatase 1 regulatory subunit 35) was shown to act upstream of CEP295 in the CCC [159,160]. PPP1R35 is a putative regulator of the protein phosphatase PP1, although its role in the CCC appears to be independent of PP1 [159]. Conceivably, CEP295, in cooperation with additional proteins, enables the formation of the inner PCM layer, which contains both CEP192 and its major functional partner, the key mitotic serine/threonine kinase PLK1 [134,141,161,162,163] (Figure 3 and Figure 5, highlighted by a purple line). Although this layer is often considered as centriole wall [6,115], it is not a part of the conserved basal body core structure because CEP192/Spd-2 is found only in Amorphea [45]. Moreover, CEP192 is a bona fide component of the PCM, and not of centrioles/basal bodies: it is present in the acentriolar mitotic MTOCs, which organize spindle poles in mouse oocytes and early embryos, but is absent from the non-centrosomal interphase MTOCs and from the basal bodies that nucleate sperm flagella, and it is not required for motile ciliogenesis [45,46,50,164,165,166,167,168]. CEP192/Spd-2 was shown to localize to the PCM-less sperm centrioles in *D. melanogaster* and *C. elegans*, but not in humans and *Xenopus laevis* [137,139,140,164,165]. In summary, centriole assembly requires CEP192 for its initiation and culminates in the recruitment of CEP192 to the outer wall of procentrioles (Figure 5).

### 4.3. Centrosome Maturation and the CCC

The CCC requires passage through mitosis, during which the newly formed procentrioles are embedded within the mitotic PCM layer (Figure 3). Thus, the CCC and centrosome maturation overlap in space and time, implying that both processes may be driven by the same mechanisms [97,99,123,124,127,134]. Indeed, the key regulator of centrosome maturation, PLK1, was shown to be essential for three key events that are integral to the CCC: (i) acquisition of the interphase PCM layer by procentrioles; (ii) disassembly of the cartwheel; (iii) centrosome disengagement at mitotic exit, which involves cleavage of PCM proteins and disintegration of the mitotic PCM layer, dissociation of the two centrosomes, and formation of the intercentrosomal linker [95,98,99,124,125,126,127,169,170] (Figure 3).

As revealed by super-resolution microscopy, the interphase PCM has an ordered, layered organization, which is conserved in from flies to humans [161,171,172,173]. Two proteins, pericentrin (PCNT) [pericentrin-like protein (PLP) in *D. melanogaster*] and CEP152/Asl, bind to the centriole wall through their C-terminus—which presumably interacts with SAS-6 and CPAP/SAS-4, respectively—and form radial fibers (with the N-terminus of PCNT and CEP152 directed outward) that follow the nine-fold centriole symmetry. Around these fibers, other PCM proteins, such as CEP192, CEP215 (centrosomin, or Cnn, in *D. melanogaster*), γ-tubulin, developmentally down-regulated protein 1 (NEDD1), and PLK1 and its activating serine/threonine kinase, Aurora A (AurA), are localized in toroidal domains [115,134,141,146,161,171,172,173,174]. The ultimate component of the PCM is γ-tubulin, which, in cells, associates with each of several γ-tubulin complex proteins (GCPs). The resulting γ-tubulin-GCP heterodimers interact laterally to form γ-tubulin complexes [3,4,28,89,175]. There are two main types of γ-tubulin complexes: a heterotetrameric γ-tubulin small complex (γ-TuSC), which is conserved throughout eukaryotes and is composed of two molecules of γ-tubulin and one molecule each of GCP2 and GCP3, and a multimeric γ-tubulin ring complex (γ-TuRC), which is found in most plants and animals, and usually consists of several laterally associated molecules of γ-TuSCs assembled together with GCP4, GCP5, and GCP6 in heterodimers with γ-tubulin [3,4,175]. In addition to these core components, the γ-TuRC may contain the adaptor protein NEDD1 and accessory proteins, such as the mitotic-spindle organizing protein associated with a ring of γ-tubulin 1 (MOZART1, or MZT1), MZT2, and the non-metastatic cells 7 protein (NME7) [176,177,178,179,180]. Variations in the compositions of GCPs can occur within both γ-TuSCs and γ-TuRCs [4,181]. As its name implies, the γ-TuRC forms a single-turn helical ring with a geometry resembling that of a 13-protofilament MT and may, therefore, act as a template for MT nucleation [3,4,175,182]. Unlike the interphase PCM, the mitotic PCM does not have a layered organization and represents a mesh-like matrix of proteins, including PCNT, CEP192, CEP215, γ-tubulin, NEDD1, AurA, and PLK1 [115,161,171,172,173].

Despite the overall conservation of the PCM layers, the functional organization of the PCM and the mechanisms underlying centrosome maturation and MT nucleation differ between animal lineages. These mechanisms have been partially delineated in *D. melanogaster, C. elegans, X. laevis,* and humans, but the picture is still incomplete, and the findings in different systems need to be reconciled [9,134,141,165,183,184,185,186]. Most of the studies have been focused on PCNT and CEP215 because these proteins and their orthologs contain an evolutionary conserved centrosomin motif 1 (CM1), which was shown to bind γ-tubulin complexes and to promote MT nucleation in different systems, including the acentriolar centrosomes of yeast [35,89,167,187]. Yeast orthologs of PCNT and CEP215 directly bind γ-tubulin complexes through the CM1 domain and recruit them to the nuclear and cytoplasmic sides of the spindle pole body, respectively (reviewed in [89,91]). Studies in *Drosophila* showed that CEP215/Cnn, in addition to its role in the CM1-mediated docking of γ-TuRC, also plays a major role in organizing the mitotic PCM layer. Phosphorylation by PLK1 promotes CEP215/Cnn oligomerization, resulting in the formation of a scaffold, which may recruit other PCM components, such as CEP192/Spd-2, PCNT/PLP, and γ-TuRC [9,183,186].

In mammalian cells, CEP215 forms complexes with PCNT, and both proteins cooperate in recruiting each other to centrosomes [188,189,190] (Figure 5, middle panel). It was proposed that CEP215-PCNT complexes form scaffolds for the recruitment of other PCM components during centrosome maturation. Notably, the interaction between CEP215 with γ-TuRC appears to be dispensable for centrosome maturation in human cells [190], suggesting the existence of a CEP215-independent mechanism(s) of centrosomal γ-TuRC recruitment and MT nucleation in mammals.

Indeed, in human cells, CEP192 is the most essential of centrosomal proteins for centrosome maturation and MT nucleation, both in mitosis and interphase [48,49,191,192]. Studies in *Xenopus* egg extracts and human cells have revealed unique scaffolding properties of CEP192, consistent with the central role of this protein in centrosome evolution and biogenesis [40,46,134,141,165]. Specifically, it was shown that CEP192 organizes AurA and PLK1 into a multistep kinase cascade, which drives the recruitment of NEDD1-γ-TuRC complexes and other PCM proteins and the consequent MT nucleation and anchoring [134]. The cascade is initiated in G2 phase, as the result of recruitment to centrosomes of CEP192, in a complex with AurA and PLK1 (Figure 5, right panel, and Figure 6A). The recruitment is likely mediated by the PLK1-phosphorylated PCNT, although this needs to be experimentally confirmed [134,193,194]. The local accumulation of CEP192 complexes in the PCM promotes oligomerization-dependent AurA activation through trans-autophosphorylation. AurA then activates PLK1 by phosphorylating it in the T-loop [134,141]. PLK1, in turn, phosphorylates CEP192 to generate multiple binding sites for NEDD1-γ-TuRC [134]. Following NEDD1-γ-TuRC recruitment, NEDD1 undergoes phosphorylation in CEP192 complexes, but the mechanism and role of the phosphorylation are unclear [134,195]. Active CEP192 complexes also promote the recruitment of the cytoskeleton-associated protein 5 (CKAP5) [also known as hepatic tumor overexpressed protein (ch-TOG) and *Xenopus* MT-associated protein of 215 kDa (XMAP215)], an MT polymerase that aids γ-TuRC in MT nucleation [196,197,198], and of other proteins [134]. The CEP192-organized kinase cascade is integral to the centrosome cycle, as it is essential for centrosome maturation and separation and for bipolar spindle assembly [134,141].

Notably, CEP192 drives γ-TuRC recruitment and MT nucleation through a mechanism, which is completely different from that described for CEP215 and PCNT. First, unlike CEP215, which interacts with γ-TuRC through the CM1 domain directly and independently of NEDD1 [89,187,199], CEP192 recruits γ-TuRC in a complex with NEDD1 in a stoichiometric ratio, suggesting that CEP192 binds pre-assembled NEDD1-γ-TuRC complexes [134] (Figure 6C,D). Second, CEP192 complexes do not directly bind NEDD1-γ-TuRC: the binding requires PLK1 docking to and phosphorylation of CEP192 [134] (Figure 6C). CEP192 complexes promote MT nucleation only when they are locally concentrated in the PCM, and the AurA-PLK1 cascade is initiated [134,165]. Third, the recruitment of CEP192 complexes is inherently coupled to the oligomerization-dependent AurA activation and the initiation of the AurA-PLK1 cascade, and these processes are facilitated by multiple feedback loops, indicative of an autocatalytic process of AurA activation and PCM protein recruitment. Indeed, the AurA-PLK1 cascade culminating in MTOC formation was recapitulated by artificial clustering of CEP192 complexes on beads coated with a recombinant N-terminal fragment (amino acids 1–1000) of CEP192 (Figure 6B) or with a bivalent anti-AurA antibody in *Xenopus* egg extracts [134,165]. Fourth, the CEP192-organized kinase cascade drives the recruitment not only of NEDD1-γ-TuRC but also of other proteins, which may be involved in different centrosome-associated processes [134,200]. The CEP192-organized kinase cascade comprises multiple steps and likely involves other proteins, in addition to the core CEP192-AurA-PLK1 complex. These proteins may participate in the recruitment, oligomerization, and posttranslational modifications of CEP192 complexes and in the docking of PLK1, NEDD1-γ-TuRC, and other proteins [134,165].

Thus, centrosome maturation in vertebrates appears to rely on two distinct, but cooperative pathways, or modules: the PCNT-CEP215 module, which promotes the assembly of the mitotic PCM scaffold, and the PCNT-CEP192 module, which drives autocatalytic recruitment of NEDD1-γ-TuRC and other PCM proteins and MT nucleation and anchoring [134,141,165,190,193] (Figure 5 and Figure 6A). The former module may have emerged earlier in evolution than the latter one because an ortholog of CEP215 appears to be present in the genome of *G. lamblia*, whereas CEP192 is only found in Amorphea [45,46]. It can be inferred that the increasing complexity of mitosis, along with the expanding role of the Aurora-PLK1-mediated signaling, provided a selective force for the evolution of the CEP192-mediated autocatalytic mechanism of PCM protein recruitment [200]. In vertebrates, this mechanism may have taken over the less efficient mechanism of recruitment of γ-tubulin complexes through direct binding to PCNT and CEP215, which is used in yeast and early-branching animals [89,91]. In addition, in the vertebrates, PCNT may have acquired the ability to organize the outer, mitotic PCM layer by cooperating with both CEP215 and CEP192 (Figure 5). Indeed, in *Drosophila*, unlike in vertebrates, PCNT/PLP does not play a significant role in the formation of the outer PCM layer [115,201,202], which could explain why in *Drosophila*, the mitotic PCM scaffold is organized by oligomers of CEP215/Cnn and not by PCNT-CEP215 complexes, as in mammalian cells [183,186,190]. As a trade-off, PCNT may have lost the ability to bind γ-TuRC directly, as suggested by the degeneration of the CM1 domain of PCNT in animals [89]. Furthermore, it is tempting to speculate that CEP192 and PLK1 of the inner PCM layer proximal to the centriole wall [115,161,193] (Figure 3 and Figure 5, highlighted by a purple line) drive PCM protein recruitment and assembly around the procentriole in mitosis.

## 5. Centrosomes in Proliferating Animal Cells

As noted above, basal bodies originally evolved to form motile cilia and flagella-organelles, which are widespread across eukaryotes, irrespectively of whether centrosomes are present or not [40,68,92]. This fact might have been the reason for a common assumption that centrosomes and primary cilia are distinct organelles with different functions and that centrioles must be converted to basal bodies in order to form primary cilia [100,203,204]. It was thought that the conversion occurs only in cells exiting the cell cycle and entering a quiescent or differentiated state. In a more extreme view, primary ciliogenesis and cell cycle progression were seen as mutually exclusive events (reviewed in [205,206]). Accordingly, it was often assumed that the roles of centrioles as platforms for the assembly of centrosomes and of primary cilia are mutually exclusive and that primary cilia have to be disassembled prior to mitosis to enable centrosome functions at mitotic spindle poles [6,203,207,208,209,210]. Accumulating evidence challenges these assumptions and suggests that primary ciliogenesis not only occurs during cell proliferation but is also a common property of most proliferating animal cells [205,211]. Studies using artificial synchronization protocols revealed a wave of ciliation in cultured mouse NIH 3T3 fibroblasts and human retinal pigment epithelial (RPE1) cells, wherein primary cilia assembled in G1 phase and disassembled in S phase [208,212,213]. By contrast, a previous study by Rieder and colleagues showed that the resorption of primary cilia in rat kangaroo kidney epithelial (PtK1) cells occurs during early mitosis (prophase and prometaphase) [214]. In agreement with this latter study, a recent work using a biosensor that allows simultaneous live imaging of the cell and cilia cycles revealed that, in proliferating cultured NIH 3T3 cells and in various cells of the mouse embryo, primary cilia assemble mostly during G1 phase and disassemble at the end of mitosis [211]. In keeping with this finding, it was shown that the anaphase-promoting complex/cyclosome (APC/C), an E3 ubiquitin ligase, which drives mitotic exit, promotes ciliary resorption, as well as centrosome disengagement [98,215]. These observations are consistent with studies showing that centrioles can nucleate primary cilia and participate in spindle pole assembly at the same time. In dividing embryonic neocortical stem cells and transformed human embryonic kidney (HEK293) cells, the primary cilium does not disassemble completely and the ciliary membrane remnant persists through mitosis at one spindle pole (while being attached to the mother centriole), acting as a determinant for the temporal and spatial control of ciliogenesis in the daughter cells [216]. Furthermore, in insect spermatocytes, centrioles promote the assembly of primary cilia and retain cilia in internalized sheaths of the plasma membrane associated with meiotic spindle poles [217,218]. A similar phenomenon was also observed in flagellated protists [217], suggesting that it is an ancestral trait.

Two important conclusions can be drawn from these studies. First, the primary ciliogenesis occurs during normal cell cycle progression in G1-S phases. Although in some cells, primary cilia may need to be disassembled by the end of mitosis to prevent a G1/S arrest (the mechanism of which is incompletely understood), in certain cell types, the ciliary membrane may persist through mitosis [207,216,219,220]. Second, the primary cilia are assembled not by basal bodies, as the traditional view holds, but by centrioles of the mother centrosomes that have relocated from the nuclear membrane to the plasma membrane. Indeed, the centrioles that form primary cilia recruit PCM proteins, such as PCNT, AurA, and γ-tubulin, indicating that the centrioles are surrounded by the PCM [208,221,222,223,224]. Furthermore, the fact that primary cilia assemble in G1/S and disassemble in late mitosis [211] implies that primary cilia or their parts are associated with centrosomes during both procentriole assembly and centrosome maturation. Since—as discussed above—procentriole assembly and centrosome maturation require the PCM and can occur only in the context of centrosomes (i.e., centrioles that have been converted to centrosomes in the previous cell cycle), it is implicit that the centriole, which assembles a primary cilium, is a part of the centrosome in proliferating cells. Remarkably, this conclusion—that the animal centrosome and the plasma-membrane-associated centriole complex that nucleates the primary cilium are essentially the same organelle—is the very definition of the Henneguy–Lenhossek hypothesis, which was formulated over a century ago and which is nowadays used in a narrower sense, to refer to the interconvertibility of centrioles and basal bodies [217].

Thus, in proliferating cells, centrosomes change their role between the juxtanuclear MTOCs and the plasma membrane-associated primary cilium organizers (Figure 1G and Figure 7). These oscillations in centrosome localization and function may serve at least two purposes in proliferating cells. First, the primary cilium may guide cell fate decisions and tissue morphogenesis by controlling spindle orientation (which defines the plane of cell division) and the distribution of cell fate determinants between the two daughter cells [216,225,226,227]. Second, the primary ciliogenesis may allow integration of extracellular cues with intracellular responses mediated through the cytoskeleton, cell division machinery, and transcription machinery. Indeed, while centrosomes are the major hubs for intracellular signaling [12], primary cilia serve as receivers and transmitters of extracellular signals. To this end, primary cilia accumulate various signaling and regulatory molecules, including ion channels, cell membrane receptors, and transcription factors [210,228,229,230]. Thus, the oscillations between the two roles of centrosomes can be seen as transitions between the extracellular (non-cell-autonomous) and intracellular (cell-autonomous) signaling modes, respectively (Figure 7).

Notably, however, not all proliferating cells form primary cilia. It was suggested that primary ciliogenesis requires polarity cues and cell adhesion or cell-cell contacts and may be inhibited in spherical cells in suspension [231].

Indeed, primary cilia are not found in circulating lymphocytes and granulocytes [205,217,231]. Primary cilia are also frequently absent in cancer cells, likely as a reflection of cell-signaling derangement [232,233,234]. In cytotoxic T lymphocytes and certain other circulating cells of the immune system, which lack primary cilia, the antigen recognition by specific receptors (such as T-cell receptors) promotes actin-dependent relocalization of centrosomes to the plasma membrane, a process called centrosome polarization. The mother centrosome in cytotoxic T lymphocytes then docks to the plasma membrane through its distal appendages and promotes the formation of the immune synapse, a structure through which cytokines and lytic granules are secreted towards target cells [15,235]. The striking similarity between the immune synapse formation and the early stages of primary ciliogenesis has led to the speculation that the immune synapse is a repurposed primary cilium [15,236]. Notably, the formation of immune synapses by CD8+ T cells promotes cell proliferation followed by differentiation into cytotoxic T lymphocytes, implying that centrosomes are fully functional and capable of returning to their roles as juxtanuclear and spindle-organizing MTOCs [15], like in the primary cilia-forming proliferating cells.

## 6. Centrosomes in Migrating Animal Cells

Cell migration is essential for tissue and organ morphogenesis during development, for tissue repair and regeneration, and for immune surveillance. It also underlies tumor dissemination [237,238,239]. In most animal cells, migration relies on dynamic changes of the actin cytoskeleton and of MTs anchored with their minus ends at the centrosome and/or the Golgi apparatus. Coordinated crosstalk between the actin and MT cytoskeletons also plays a fundamental role in cleavage furrow formation during cell division [2,240]. In animals, the actin and tubulin cytoskeletons have undergone a substantial evolution and the centrosome became a primary organizer and integrator of both cytoskeletons [40,68,241,242,243].

Cell migration is a cycling, multistep process, which includes polarization, formation of membrane protrusions and adhesions at the leading edge, and de-adhesion and retraction at the rear of the cell. These steps are continuously repeated during the migration process [238]. The migratory response critically depends on the integration of various extracellular cues into a precise control of the signaling circuits that govern the cytoskeleton. Cell polarization is a keystone of migration and is usually determined by the positioning of the centrosome relative to the nucleus (nucleus-centrosome axis). The centrosomes play a major role in polarized cell migration, and, in most cell types, they are positioned between the nucleus and the leading edge, with the majority of centrosomal MTs being directed towards the leading edge [237,238,244,245]. The mother centrosome in migrating cells may also form a primary cilium, which orients along the axis of migration [246,247]. The involvement of the primary cilia in cell migration is supported by the observations that many signaling pathways transduced through primary cilia—such as those involving sonic hedgehog (SHH), wingless-type MMTV integration site family (WNT), transforming growth factor β (TGFβ), receptor tyrosine kinases [most notably, platelet-derived growth factor receptor, α polypeptide (PDGFRα)], and G-protein coupled receptors—control cell migration [230,247,248]. Furthermore, studies of several genetic conditions with defective cilia (ciliopathies), such as the Bardet–Biedl syndrome, the Joubert syndrome, and the Meckel–Gruber syndrome, have implicated primary cilia in cell migration during brain development [247,249].

Evidence suggests that the primary cilium may act as an antenna and a cellular global positioning system (GPS), which detects and transmits chemical and mechanical cues from the outer milieu to the cell [247,250,251]. The mother centrosome may then transduce and integrate the effector signals that impinge on the modulation of the dynamics of the actin and MT cytoskeletons, of focal adhesions, and of trafficking along centrosomal MTs [247]. In certain cell types, such as the corneal endothelial cells and the inner ear hair cells, primary cilia are formed during development and tissue repair but are disassembled during steady state in normal adult tissues [252,253]. The primary cilia reassemble in response to an injury [252]. In tangentially migrating neurons, the centrosome acts both as an MTOC and a primary cilia organizer, and may also gather in the same compartment as the Golgi apparatus, which serves as an MTOC in its own right [247,254,255,256]. During the migration cycle of these neurons, the centrosome changes its position along the front-back axis and also oscillates between the nuclear membrane and the plasma membrane. Moreover, the mother centrosome does not permanently dock to the plasma membrane; rather, the primary cilium is repeatedly formed and removed from the cell surface by fusion/fission of the ciliary vesicle [254]. These findings suggest that the oscillations between the two states of the centrosome (juxtanuclear MTOC and primary cilium organizer) occur not only in proliferating cells (Figure 7) but are also integral to the cell migration cycle. The role of such oscillations may be different during cell proliferation and migration. It was suggested that the dynamics of formation, orientation, and length of the primary cilium serve as switches to control the migratory response [247]. Further studies are needed to test these hypotheses and elucidate the underlying mechanisms.

## 7. Centrosomes in Postmitotic Differentiated Cells

Centrosomes are thought to be primarily cell division organelles with an ancestral role in spindle assembly [6,10,42,91]. Indeed, despite the existence of several redundant spindle assembly pathways, centrosomes are essential for proper chromosome segregation: centrosome ablation prolongs mitosis and causes a postmitotic, p53-dependent G1 arrest in mammalian cells [257,258,259,260,261]. Another strong evidence of the primary role of centrosomes in cell division comes from the findings that the MT-nucleating capacity of centrosomes is downregulated, and sometimes is completely lost, in postmitotic differentiated cells [33,34]. However, some interphase processes, such as cell migration and immune synapse formation, require a substantial MT-nucleating capacity of centrosomes. Below, we have discussed the role of centrosomes during cell differentiation and the possible underlying mechanisms.

### 7.1. Downregulation of Centrosome Function in Differentiated Cells

Cell differentiation is usually accompanied by attenuation of MT-nucleating capacity of centrosomes due to the reduction of the amount of the PCM and its MT-nucleating and anchoring capacity. In many differentiated cells, the MTOC function is reassigned to other sites as a means to generate a unique MT cytoskeleton architecture best suited for each particular cell type. The cell cortex, the apical plasma membrane, the nuclear envelope, the Golgi apparatus, the mitochondria, and the sides of cytoplasmic MTs were shown to serve as non-centrosomal MTOCs in various differentiated cells [33,34,35]. The choice between these sites and the degree of centrosome inactivation are specific for each cell type. A complete loss of the centrosomal MTOC activity with its reassignment to non-centrosomal sites was shown to occur in myotubes (multinucleated cells formed by the fusion of differentiated myoblasts), postnatal cardiomyocytes, neurons, and certain epithelial cells [33,34,262,263,264]. The mechanisms underlying MT nucleation and anchoring at non-centrosomal MTOCs are poorly understood, although progress has been made, particularly with regards to the Golgi apparatus and mitochondria [33,34,167,255]. 

MT nucleation and anchoring at some non-centrosomal MTOCs, such as the Golgi apparatus, the nuclear envelope, and sperm mitochondria, was shown to involve PCNT and CEP215 and their isoforms and/or paralogs [35,89,167,256,262]. A recent study showed that targeting an engineered protein containing the CM1 domain from the human or *Drosophila* CEP215 to mitochondria was sufficient to convert these organelles to MTOCs [167]. These mitochondrial MTOCs recruited γ-TuRCs and NEDD1, but not any other PCM proteins, providing the first evidence that spatial targeting of the CM1 domain is sufficient to generate localized MTOCs [167]. Of note, this observation is at odds with previous studies suggesting that CEP215 recruits γ-TuRC through direct binding, independently of NEDD1 [187,199]. This contradiction needs further investigation. MT assembly at the cis-face of the Golgi apparatus relies on a pathway, which appears analogous to the centrosomal PCNT-CEP215 pathway (Figure 5, module 2), and which involves PCNT and CEP215, as well as their paralogs, A-kinase anchoring protein 450 (AKAP450) [also known as AKAP9 or centrosome- and Golgi-localized protein kinase N-associated protein (CG-NAP)] and myomegalin, respectively. AKAP450 and myomegalin are substantially more abundant at the Golgi apparatus than at centrosomes, whereas the opposite applies to PCNT and CEP215 [190,255,256]. AKAP450 accumulates at the cis-Golgi by binding to the Golgi matrix protein GM130 and recruits CEP215 and myomegalin through direct interaction. CEP215 and myomegalin promote CM1 domain-mediated γ-TuRC recruitment and the consequent MT nucleation and anchoring. Conceivably, PCNT may in part substitute for AKAP450 in this pathway [255,256,265,266]. Whether AKAP450 and PCNT also contribute to MT nucleation through direct binding of γ-TuRC is unclear because the CM1 domain is degenerate in both paralogs [89].

These observations suggest that the relocalization of the PCM proteins, in particular, of the PCNT-CEP215 module (Figure 5, module 2), contributes to MT nucleation and anchoring at multiple non-centrosomal sites. The formation of non-centrosomal MTOCs in differentiated cells is facilitated by the proteins of the CAMSAP (calmodulin-regulated spectrin-associated proteins)/Patronin family [5,267,268]. These proteins specifically bind and stabilize uncapped MT minus ends and support MT minus-end growth independently of γ-TuRC [268,269,270,271]. Such unique properties of the CAMSAP proteins are mediated by a conserved, family-defining CKK (CAMSAP1, KIAA1078 and KIAA1543) domain, which recognizes subtly specific tubulin conformations at the MT minus end [272,273,274]. Notably, the CAMSAP proteins are found only in animals with differentiated tissues, but not in any other organisms, such as those of non-animal lineages or sponges, which lack tissues [272]. Thus, the CAMSAP proteins appear to have evolved specifically to organize non-centrosomal MT networks in differentiated cells.

### 7.2. Primary Cilia in Differentiated Cells

Given the attenuation of centrosome function during cell differentiation and the long-known reciprocal relationship between the formation of juxtanuclear centrosomes and primary cilia [223,275,276], it is not surprising that most differentiated, quiescent cells form primary cilia [277]. Indeed, all mammalian tissues contain populations of ciliated cells, although there are examples of cell types—including those of epithelial origin—which lack primary cilia in situ [205]. The choice on whether to form a primary cilium or a juxtanuclear centrosome depends on the cell type, the developmental stage of the organisms, and the environmental conditions. The primary cilia were suggested to have different roles in differentiated cells, including the maintenance of growth arrest and of differentiated state, the preservation of regenerative capacity, the suppression of cyst formation and of oncogenesis, and the assistance in cell migration [205,277]. Furthermore, several differentiated tissues use primary cilia for highly specialized functions. The most notable examples are the sensory organs, like the inner ear, the nose, and the eye, in which the role of the primary cilium as a sensory organelle has been exploited to the fullest extent [221,278].

In the inner ear of vertebrates, hair cells are the sensory receptors that detect and convert sound and head motion into signals that are interpreted by the brain [221,279]. Each hair cell has a non-motile primary cilium, called kinocilium, with a (9 × 2 + 2) MT configuration (normally found in motile cilia [Figure 4]), along with several actin-filled microvilli. The kinocilium is essential for the differentiation of hair cells, and therefore for the hearing process, although it is itself not involved in sound transduction and degenerates after birth [221]. In the nasal epithelium, the olfactory sensory neurons detect and transmit odorant information to the central nervous system. They have 10 to 30 non-motile primary cilia with a (9 × 2 + 2) MT configuration, which are formed by mother centrioles of centrosomes generated through centrosome amplification [221]. These cilia enable the perception of smell through the massive family (~400 members in humans and >1000 members in mice) of G protein-coupled olfactory receptors, or odorant receptors, which localize to the ciliary membrane [221,280]. It is generally accepted that each olfactory sensory neuron expresses one and only one olfactory receptor [281,282,283]. Binding of odorants or mixtures of odorants to a specific pattern of odorant receptors initiates a cAMP-dependent signaling cascade, which amplifies and transmits signals to the brain, causing the sensation of smell. All elements of the cascade are enriched in the olfactory sensory cilia, underscoring the key role of these organelles in olfaction [221,284]. In the vertebrate retina, rods and cones are specialized neurons optimized for the detection of light and are, therefore, called photoreceptors. They possess a highly modified primary cilium, the axonemal part of which (outer segment) is filled with stacks of coin-shaped membranes containing light-sensitive pigment rhodopsin at a concentration that reaches the highest level for known membrane proteins [228,285]. As with the olfactory sensory neurons, the signal transduction machinery—in this case, for light detection—is also localized to the primary cilium (its outer segment) of rod photoreceptors. This unique ciliary organization endows rod photoreceptors with the astonishing ability to respond reliably to single photons [221,228,285,286].

Notably, in the olfactory sensory neurons and photoreceptors, the centrioles that form primary cilia are surrounded by the PCM, which contains PCNT and nucleates MTs [221,284,287]. PCNT is required for the assembly of the olfactory sensory cilia [222,288]. Furthermore, in photoreceptors, MTs nucleated by the PCM serve as tracks for the dynein 1-mediated retrograde transport of rhodopsin from the Golgi apparatus located in the inner segment of the cell. Rhodopsin and other components of the phototransduction cascade are then delivered from the centrosome to the outer segment through anterograde intraflagellar transport and myosin-driven transport along actin filaments (reviewed in [221]). These findings exemplify two types of differentiated cells (i.e., olfactory sensory neurons and photoreceptors) in which the mother centrosome organizes a highly specialized primary cilium and, at the same time, serves as an MTOC that is also involved in the sensation process. This is yet another evidence that the roles of the centrosome as an MTOC and a primary cilium organizer are non-mutually exclusive.

### 7.3. Mechanisms of Centrosome Inactivation During Cell Differentiation

It has been shown that different MTOCs may reciprocally regulate each other through competition for MT assembly factors. As such, centrosomes are dominant MTOCs not only with regards to the extent of their own MT-nucleating activity but also because this activity may indirectly (through consumption of MT-nucleating factors) suppress the formation of non-centrosomal MTOCs [168,191,192,289]. Accordingly, as centrosomes are gradually inactivated during differentiation, their MTOC function is attenuated and reassigned to non-centrosomal sites. Hence, the deciphering of the mechanisms underlying centrosome inactivation is key to understanding how the cytoskeleton is remodeled during differentiation.

The activity of centrosomes as MTOCs is maximal in mitosis, consistent with the notion that centrosomes are, ancestrally, cell division organelles. Accordingly, PLK1—a mitotic kinase expressed at high levels in G2-M phases—along with CEP192, is essential for centrosome biogenesis and is thought to have played a key role in the evolution of the animal centrosome [40,93,162,200]. Before the onset of mitosis, PLK1 is activated in the cytoplasm by AurA, in a complex with the scaffold protein Bora [290,291]. PLK1 activation at mitotic signaling platforms is ensured by two other complexes: the CEP192 complex activates PLK1 at centrosomes, whereas its counterpart, the chromosomal passenger complex—which is organized by the inner centromere protein (INCENP) and uses Aurora B as a catalytic subunit—activates PLK1 at centromeres, kinetochores, and the midbody [58,134,141,200,292,293,294,295]. During mitosis, PLK1 docks onto a myriad of proteins through its C-terminally-located Polo-box domain [296,297,298]. The Aurora-PLK1 scaffolding proteins and the numerous PLK1-docking proteins work together and with the upstream regulator, CDK1^Cyclin B^ complex, to control mitosis in space and time [200,299,300]. Mitotic entry and exit are switch-like transitions that are driven by the conserved mitotic oscillator comprising CDK1^Cyclin B^ and its antagonist, APC/C^Cdc20^ [301,302,303,304,305]. CDK1 is the most important kinase of its family because, in the absence of other CDKs, it alone is sufficient to drive the cell cycle in mammalian cells [306]. Activation of CDK1^Cyclin B^ above a certain threshold sets up commitment to mitosis, after which the cell cannot return to interphase [300,305]. 

A number of studies in different organisms have revealed that the activity of the CDK1^Cyclin B^-APC/C^Cdc20^ oscillator, when calibrated below the mitotic commitment threshold, may function as a rheostat (rather than a switch), and may be used by cells to drive terminal differentiation programs, while avoiding nuclear divisions [93,168,199,307,308,309,310,311,312]. These studies have also provided important insights into centrosome biology. Experimental lowering of the levels of mitotic cyclins in the early *Drosophila* embryo resulted in the uncoupling of the nuclear and centrosome cycle and allowed centrosome duplication in the absence of mitoses [310,312]. A natural attenuated CDK1^Cyclin B^-APC/C^Cdc20^ oscillator was shown to drive a terminal differentiation program of the mouse brain multiciliated cells, whereby multiple basal bodies for nucleating motile cilia are generated simultaneously and in an orderly fashion through the deuterosome pathway (see below–Section 10) [308]. Similarly, *Drosophila* oocytes attenuate the activity of PLK1 to downregulate PCM formation and eliminate centrosomes—an essential event in the oocyte differentiation program in animals. Ectopic tethering of PLK1 to the oocytes’ centrioles prevented centrosome loss and interfered with meiotic and mitotic divisions, leading to female sterility [93].

These findings are consistent with a recent study by Muroyama et al. suggesting that centrosome inactivation in differentiating mouse keratinocytes is driven by the attenuation of CDK1 activity during the cell cycle exit and not by the differentiation program per se [199]. This study revealed two steps of centrosome inactivation defined by the dynamics of two distinct γ-TuRC complexes. In the first step, occurring upon the cell cycle exit, centrosomes lose NEDD1-γ-TuRC complexes while retaining CEP215-γ-TuRC complexes, which results in a dramatic, but incomplete, reduction of centrosomal MT nucleation. In the second step, centrosomes lose CEP215-γ-TuRC, which completes centrosome inactivation [199]. Similarly, the loss of NEDD1 was implicated in centrosome inactivation in the differentiating rodent hippocampal neurons [263]. On the basis of these findings, it was suggested that the main role of CEP215-γ-TuRC is to nucleate MTs, whereas that of NEDD1-γ-TuRC is to anchor MTs at centrosomes [199,313]. It should be noted, however, that NEDD1-γ-TuRC complexes are present in the cytoplasm and are essential for MT nucleation in all pathways of mitotic spindle assembly [176,177]. NEDD1-γ-TuRC is recruited to and localized at different MTOCs by distinct anchoring factors. NEDD1 is, therefore, a universal adaptor—rather than an anchor—of the γ-TuRC. NEDD1-γ-TuRC is localized to the vicinity of mitotic chromatin by the RHAMM-TPX2 (receptor for hyaluronan-mediated motility-targeting protein for Xklp2) complex (chromatin-driven spindle assembly pathway) [314] and is anchored at MT walls by the multisubunit Augmin complex (MT-driven spindle assembly pathway) [315,316,317]. NEDD1-γ-TuRC also localizes to the central spindle through an unknown mechanism [318]. As noted above, NEDD1-γ-TuRC is recruited to and anchored at centrosomes in G2-M phases by active CEP192 complexes, although it is yet unclear whether the docking of NEDD1-γ-TuRC to the PLK1-phosphorylated CEP192 is direct or involves additional factors [134] (Figure 6). It is, therefore, conceivable that in the aforementioned study by Muroyama et al., the first step of centrosome inactivation in keratinocytes, which involves the loss of NEDD1-γ-TuRC from the PCM [199], comprises switching off the PCNT-CEP192-mediated mechanism of MT nucleation with retention of the PCNT-CEP215 mechanism. In support of this notion, cell-fusion experiments in *C. elegans* have revealed that the downregulation of mitotic CDK and of CEP192/Spd-2 underlies centrosome inactivation and the reassignment of the MTOC function to the apical plasma membrane during differentiation of embryonic intestinal cells. The centrosomal MTOCs can be rapidly reactivated in quiescent or differentiated cells by supplying CDK and CEP192/Spd-2 from mitotic cells [168]. Furthermore, the first step of centrosome inactivation in *C. elegans* was shown to involve a protein phosphatase PP2A-mediated removal of CEP192/Spd-2 from centrosomes [319,320], consistent with the strict dependence of the PCNT-CEP192 module on protein phosphorylations in the vertebrates [134,165,193].

Together, these studies imply that centrosome inactivation in differentiating cells is driven by attenuation of the activities of the CDK1^Cyclin B^-APC/C^Cdc20^ oscillator and, consequently, of CEP192/Spd-2 and PLK1, which are the core components of the MT-nucleating CEP192 complex involved in centrosome maturation [134,141]. Below, we have explored this notion with regards to the mechanisms of centrosomal MT nucleation in interphase cells.

### 7.4. Model of Centrosomal MT Nucleation in Interphase and Differentiating Cells

The aforementioned studies suggest that the dynamics of centrosome remodeling and MT nucleation during cell differentiation is inverse to that during cell cycle re-entry of quiescent or differentiated cells (Figure 8). The findings [93,168,191,199], taken in the context of previous studies [49,134,187,188,190,192,193], also imply that, as in mitosis, there are two modes of centrosomal MT nucleation and anchoring in interphase: the one mediated by the PCNT-CEP215 module and the other one mediated by the PCNT-CEP192 module (Figure 5). Because CEP215 binds PCNT and γ-TuRC directly and independently of other factors or posttranslational modifications [187,190], the PCNT-CEP215 module is less dependent on PLK1 activity but is also less efficient in MT nucleation than the PCNT-CEP192 module, which drives NEDD1-γ-TuRC recruitment in a phosphorylation-dependent, autocatalytic manner. Therefore, the PCNT-CEP215 module may operate throughout the cell cycle, ensuring centrosomal MT nucleation at a basal level (Figure 8). A minimal PLK1 activity may still be required for centrosome functions in interphase because complete PLK1 inactivation results in centrosome loss in *Drosophila* [93].

Unlike the PCNT-CEP215 module, the PCNT-CEP192 module is absolutely dependent on PLK1 activity, because PLK1 phosphorylation of PCNT presumably initiates the recruitment of CEP192 complexes to centrosomes (which, in turn, initiates the AurA-PLK1 cascade), and because PLK1 docking to and phosphorylation of CEP192 is a prerequisite for NEDD1-γ-TuRC recruitment and MT nucleation [134,193] (Figure 6C,D). Since the PCNT-CEP192 module enables centrosome maturation prior to mitotic commitment, in G2 phase, it is conceivable that this module may also promote MT nucleation in interphase and quiescent cells, if PLK1 activity is high enough to drive the PCNT-CEP192 recruitment (Figure 8). Several lines of evidence support this hypothesis. First, in mammalian cells, CEP192 is essential for centrosomal MT nucleation both in mitosis and interphase [48,49,191,192]. Second, the level of cytoplasmic CEP192 determines the size and MT-nucleating activity of centrosomes both in mitotic and interphase cells [163,168,191,192].

Third, anti-AurA antibody-coated beads, which are known to promote MT nucleation through the CEP192-organized kinase cascade, form large MT asters during metaphase-to-interphase progression in *Xenopus* egg extracts [134,165,321,322]. Fourth, because AurA is a catalytic subunit of the CEP192 complex, the activation of PLK1 in this complex can be uncoupled from and precede that of the main pool of PLK1 in the cytoplasm, which is activated by Bora-AurA [134,200]. This may explain why the centrosome cycle can be uncoupled from the nuclear cycle [310,312]. In fact, the main role of AurA in CEP192 complexes may be to aid centrosome functions outside mitosis, when the cytoplasmic PLK1 is not fully active [200]. In addition, experiments in *Xenopus* egg extracts revealed that the level of NEDD1-γ-TuRC recruitment and MT nucleation by the CEP192 complex is correlated with the number of PLK1-phosphorylated serines in CEP192 [134] (Figure 6D), suggesting that the CEP192 complex may act as a rheostat that regulates MTOC function depending on the activities of CDK1 and PLK1.

Thus, the function of centrosomes as MTOCs in interphase cells may be regulated by a dual-circuit mechanism involving the PCNT-CEP215 module and the PCNT-CEP192 module. Such a mechanism may enable a wide range of MT-nucleating capacities of centrosomes that can be calibrated by the CDK1^Cyclin B^-APC/C^Cdc20^ oscillator to optimally suit each differentiation program and functional state of the cell (Figure 8). Other factors, such as protein abundance, may modulate the activity of each module. Indeed, CEP192 and PCNT were shown to maintain an antagonistic relationship at interphase centrosomes, with CEP192 suppressing the centrosomal accumulation of PCNT and promoting MT nucleation, and with PCNT inhibiting the centrosomal localization of CEP192 and PLK1 and MT nucleation [192,323]. This observation is consistent with our model (Figure 8). Because CEP192 is expected to bind only the PLK1-phosphorylated PCNT (which may be scarce in the interphase PCM), excess of unphosphorylated PCNT may dilute or hide the CEP192-docking sites on PCNT, thereby shifting the balance towards the less efficient PCNT-CEP215 pathway of MT nucleation. Conversely, lowering the concentration of PCNT may increase the stoichiometry of PCNT phosphorylation by PLK1 in the interphase PCM, fostering the CEP192-mediated, autocatalytic mechanism of NEDD1-γ-TuRC recruitment. Hence, the PCNT-CEP192 module appears to provide a second level of regulation of the interphase centrosomes, which may be important for the function of these organelles in such processes as cell polarization, migration, and immunological synapse formation, which require robust centrosomal MT nucleation [15,237,247]. Indeed, CEP192 was shown to be required for efficient polarization and cell migration of human osteosarcoma U2OS cells [192].

## 8. The Centrosome-Connecting System

From the above, it follows that the animal centrosome is a composite organelle capable of acting as an MTOC and a signaling platform at either of the two ends of the ancestral nucleus-basal body connector–the nuclear membrane or the plasma membrane (Figure 1 and Figure 2). The duality of the animal centrosome (Table 1) may have provided a selective force for the evolution of a new, rootletin/C-NAP1-based fibrous cytoskeletal system, which enables both centrosome cohesion and association of the primary cilium with the cellular interior. Indeed, rootletin is the main constituent of not only the intercentrosomal linker but also of the ciliary rootlet (Figure 3 and Figure 4), a bundle of fibers, which project from the proximal end of the cilium-forming mother centriole towards the cellular interior and may connect to the nuclear envelope, the Golgi apparatus, and, possibly, other organelles [104,105,324,325,326,327]. Notably, the rootletin filaments dock at the nuclear surface through the linker of the nucleus and cytoskeleton (LINC) complexes, which are composed of the Sad1 and UNC-84 (SUN) domain proteins and Klarsicht, ANC-1, and Syne homology (KASH) domain proteins located at the inner and outer nuclear membranes, respectively, and which are also involved in the centrosome-to-nucleus connection [328,329,330,331,332]. Conceivably, the rootletin filaments have evolved as a substitute for the contractile centrin fibers and the non-contractile striated fiber assemblin (SFA) fibers, which form the nucleus-basal body connector in lower eukaryotes. This evolutionary substitution suggests that the connection between the centrioles/basal bodies and the nucleus and the Golgi apparatus is a universally important ancestral trait in eukaryotes [37,72,75,76,77,333].

Evidence suggests that the intercentrosomal linker is formed as the result of interdigitation and entanglement of multiple, highly stable rootletin filaments emanating from the proximal end of each parental centriole [102,130,131]. These filaments gradually form during the transition from anaphase to G1 phase [131]. A recent study using super-resolution microscopy has revealed that the filaments are composed of ordered, repetitive units of rootletin, in association with CEP68, and are organized and anchored by a C-NAP1 ring and a rootletin/CEP68 ring at the proximal end of each centriole [130]. The filament assembly, like the CCC, requires licensing by PLK1 activity and passage through mitosis, implying that it is a part of the CCC program [131]. The interdigitating rootletin filament network has substantial plasticity, which may explain some aspects of the centrosome cycle. Specifically, it was proposed that the interdigitating rootletin filaments can be compacted, causing coalescence of the two centrosomes together into a single MTOC, disentangled, resulting in a transient splitting apart of the two centrosomes into separate units connected by the linker, or can be broken by the MT-sliding force generated by kinesin KIF11/Eg5 to enable centrosome separation and bipolar spindle assembly [130,131].

These findings also imply that, in ciliated cells, the mother centrosome forms two types of connecting rootletin fibers: the filaments involved in centrosome cohesion and the filaments that comprise the rootlet directed towards the cellular interior (Figure 3 and Figure 4). How these filaments differ in their composition and what factors confer the selective ability of the cilium-forming mother centriole to assemble the rootlet is unknown. There is substantial variability in the length and organization of the intercentrosomal linker and the ciliary rootlet and in the role of these structures between different organisms and cell types. The ciliary rootlet does not seem to be essential for primary ciliogenesis but is required for proper ciliary function and maintenance [325,327,334,335,336]. Experimental evidence indicates that centrosome cohesion and the coalescence of the two centrosomes into a single MTOC are important for cell migration and organization of the cytoskeleton and of the Golgi apparatus in interphase and for proper spindle assembly and chromosome segregation in mitosis [324,325,337,338,339,340]. The physical connection of the centrosome and/or the primary cilium to the cell nucleus may be important for cell migration, hearing sensation, and immunological synapse formation because these processes were shown to depend on both the centrosomes/primary cilia and the intact LINC complexes [15,341,342,343,344,345,346]. Consistent with these observations, studies in *Drosophila* have implicated ciliary rootlets in mechano- and chemo-sensation, suggesting that these structures may mediate signal transduction from the primary cilium to the nucleus [335,336].

## 9. Evolutionary Benefits of Centrosomes and Primary Cilia

It can be said that the animal centrosome is unique in that it can assemble a primary cilium (see Section 5), as the primary cilia are unique in that they can only be formed by centrosomes. Indeed, centrosome inactivation in differentiated cells owing to PCM disintegration correlates with the loss of primary cilia, even when centrioles are still present [262,347]. Furthermore, a secondary loss of CEP192/Spd-2 and four additional “centrosome signature genes” in the planarian flatworm *Schmidtea mediterranea* resulted in the loss of both centrosomes and primary cilia. Notably, the lack of primary cilia in *S. mediterranea* is due to the loss of centrosomes and not centrioles, because basal bodies are formed de novo and assemble motile cilia in multiciliated cell types in this organism [40,46]. Similarly, in multiciliated mammalian cells, the basal bodies for motile cilia are formed de novo independently of centrosomes, although this process involves certain centrosomal proteins, which function downstream of CEP192 [46,348,349,350]. In vertebrates, motile sperm flagella are formed by centrioles, which are surrounded by only a minimal amount of PCM that lacks the principal PCM proteins CEP192, γ-tubulin, PCNT, and CEP152 (as a result of PCM reduction during spermiogenesis) [164,165,351]. According to the above definition, such centrioles should be considered as basal bodies, indicating that the mature sperm lacks functional centrosomes. Thus, unlike the primary ciliogenesis, which requires centrosomes, the motile ciliogenesis occurs in the absence of the PCM, or at least of its major components. How the centrioles and the PCM work together to enable the assembly and function of the primary cilium is largely unexplored, although there is evidence of cross-talk between the two structures [221,223,247].

The juxtanuclear MTOC and the primary cilium organized by the animal centrosome are very different, if not antipodal, structures, consistent with the notion that the principal components of these structures, the PCM and the centrioles, respectively, originate from two distinct ancestral MTOCs (Table 1). Recent progress in centrosome research and in comparative genomics shed light on the evolutionary origin and benefits of the animal centrosome. Phylogenetic evidence suggests that centrosomes in different eukaryotic lineages evolved through convergent evolution and that the key event in the evolution of the animal centrosome was the emergence of CEP192/Spd-2 [40,45,46,65]. This hypothesis is supported by experimental evidence of the central role of CEP192 in PCM formation and MT nucleation, in integrating these processes with centriole assembly, and in controlling the centrosomal pool of AurA and PLK1 (and thereby linking the centrosome cycle to the mitotic regulatory network) (reviewed in [200]) (Figure 5). The phylogenetically lowest organism in which a CEP192/Spd-2 ortholog was identified is an early branching amorphean, the social amoeba *Dictyostelium discoideum* [40,45,67]. Accordingly, *D. discoideum* has a centrosome that lacks centrioles, but otherwise resembles the animal centrosome: it has a corona reminiscent of a PCM, which surrounds a three-layered core structure; it organizes mitotic astral MTs, which are required for proper cytokinesis; like the animal centrosome, it is attached to the nuclear envelope and nuclear lamina through LINC complexes [67,329,331,332]. Remarkably, *D. discoideum* stands on an evolutionary scale on the threshold between unicellular and multicellular organisms [40,67,352]. Indeed, *D. discoideum* forms multicellular structures composed of motile and differentiated cells, which resemble epithelia of animals and share key features with animal tissues, such as cell adhesion, communication, signaling, and differentiation [353,354]. As shown above, in animals, cell motility, adhesion, differentiation, and signaling, involve centrosomes. These observations imply that the transition to complex multicellularity provided selective pressure for the evolution of centrosomes in Amorphea. Another amoebozoan, *Physarum polycephalum,* forms centriolar centrosomes (among several types of MTOCs formed at different life cycle stages), which are similar to those of animal cells, and, therefore, can be considered as prototype centrosomes of Amorphea [40]. Unlike *D. discoideum*, *P. polycephalum* is a unicellular amoeba, and it seems to lack CEP192, presumably, as the result of a secondary loss (reviewed in [40,67,92]). The transition to complex multicellularity may have also provided selective pressure for the evolution of centriolar centrosomes, as suggested by the presence of both these traits in the early branching eukaryote brown algae [83,355]. These organisms have centriolar centrosomes (Figure 1C) and share several features common to animals, such as complex multicellularity, the dependence of cell polarity and morphogenesis on the communication between the MT- and actin cytoskeletons, and reliance of cell division plane orientation on the centrosome position [83,356].

All animals have centriolar centrosomes, with the only known exception being *S. mediterranea*, and, possibly, other planarians, which lack centrosomes altogether [46]. *S. mediterranea* is unique in that its embryonic development is not dependent on the highly stereotyped pattern of embryonic cleavage generated by oriented cell divisions and precise cleavage plane geometry [40,46]. Thus, the loss of centrosomes in planarians is consistent with the essential role of these organelles in the preservation of cell polarity and cell individuation—the traits imposed by complex multicellularity [10]. The hybrid, centriolar centrosomes may confer several evolutionary advantages. First, they allow maintaining centrioles through the life cycle (thereby evolutionarily preserving the centriole assembly program), while avoiding the need for energy-demanding motile ciliogenesis [357]. Because most animal cell types use actin-based ameboid motility, they do not require motile cilia, and motile ciliogenesis occurs only in selected cell types independently of centrosomes. Second, the centriolar centrosomes ensure the association of centrioles with spindle poles, which is essential for faithful centriole segregation during mitosis. Third, the centriolar centrosomes allow reducing the number of primary MTOCs in a cell to no more than two. Many unicellular eukaryotes have multiple MTOCs [38,73,86], whereas animal cells have two centrosomes, which coalesce into a single MTOC in interphase. The presence of a single interphase MTOC may be essential for proper cytoskeleton organization and function (see Section 8). Finally, and perhaps most importantly, in animals, the centriolar centrosomes enable primary ciliogenesis, which brings these organelles to an entirely new level of regulation of cellular processes.

The centriole/primary cilium module and the PCM/juxtanuclear MTOC (centrosome) module may have evolved under entirely different selective constraints. The evolution of the former module may have been linked to the expansion of cellular diversity and intercommunication, whereas that of the latter module may have been associated with the growing role of cell polarity and individuation and the increasing complexity of mitosis. Although primary cilia of different animal lineages and cell types are similarly organized and use the same evolutionary conserved intraflagellar transport system (Figure 4), they are functionally diverse organelles, which have undergone a substantial clade- and cell-type-specific expansion in the repertoire [357]. Sensory and signaling pathways associated with ciliary membranes have been modified extensively and adapted to fit the needs of each organism or cell type. Furthermore, the mechanisms by which primary cilia communicate extracellular signals into cellular responses are also different depending on the type and state of the cell. For example, in some terminally differentiated cells, primary cilia are only involved in sensation and remain permanently associated with the plasma membrane [221,358]. In such cells, centrioles may disintegrate without affecting ciliary function, as suggested by a study in *C. elegans* showing that, in certain cell types, centrioles are required for the assembly, but not for maturation, or function, of primary cilia [358]. By contrast, cell proliferation and directed migration appear to rely not only on ciliary signaling but also on the cycles of ciliary assembly/disassembly and on the oscillations of centrosome localization between the nuclear membrane and the plasma membrane. The centrosome has, therefore, become an integrator of extracellular and intracellular signals and the cytoskeleton and a switch between the non-cell autonomous and cell-autonomous signaling modes (Figure 7).

## 10. Hierarchy and Modularity of the Centrosome Biogenesis Networks

Given the complexity of the MT cytoskeleton in early-branching eukaryotes, it can be inferred that the basic mechanisms for the MTOC and ciliary assembly were already present in the last common ancestor of all eukaryotes [36,40,42,43,64,359]. In many eukaryotic lineages, the MT cytoskeleton underwent a secondary morphological simplification with a partial or complete loss of the basal body apparatus (Figure 1 and Figure 2). The emergence of the centriolar centrosome was a keystone in the evolution of the MT cytoskeleton in the Amorphea. The conserved basal body assembly module was merged with the PCM assembly module involving several newly evolved proteins, such as CEP192, CEP152, and PLK4 [45,65]. Both modules were integrated with the ciliary assembly program evolved from that of the unicellular eukaryotes [203,205,209,226,230], and with the cell cycle machinery (Figure 3 and Figure 5). As a result, a canonical centrosome cycle has evolved that ensures that each cell contains two centrosomes, which function in accordance with the cell type and cell cycle phases.

Distinct modules of the centrosome cycle may have been repurposed for de novo assembly of basal bodies or non-centrosomal MTOCs. Such a strategy may have played a key role in the diversification of the MT cytoskeleton in differentiated cells. For example, in postmitotic multiciliated cells, basal bodies are formed de novo by specialized structures termed deuterosomes [360,361]. Deuterosomes promote basal body assembly through a pathway, which is analogous to that for centriole assembly but is initiated downstream of CEP192, at the level of CEP152 (Figure 5, module 1). The pathway involves deuterosome-specific proteins CCDC78 and deuterosome assembly protein 1 (DEUP1), which is a paralog of CEP63. CCDC78 recruits CEP152-DEUP1 complexes, which, in turn, promote the recruitment of PLK4 and SAS6 to initiate basal body assembly [362,363,364]. Notably, PLK4 activity is required only for centriole duplication, but not for the deuterosome-mediated basal body assembly de novo in postmitotic multiciliated cells (although PLK4 protein itself may facilitate the latter process). These observations are consistent with the primary role of PLK4 and CEP192 in the canonical centrosome cycle and reinforce the notion that the deuterosome-mediated generation of basal bodies—and the motile ciliogenesis, in general—are independent of centrosomes [46,348,349,350]. DEUP1 and the deuterosome pathway are present only in vertebrates, and CEP152 is found only in animals, whereas the de novo basal body assembly occurs in many eukaryotic lineages [43,64,361]. Thus, multiciliogenesis and the de novo basal body assembly may involve alternative, deuterosome-independent mechanisms, which may be initiated at the level of CEP152 or downstream of it [360,361]. In multiciliated cells of *S. mediterranea* (which lacks centrosomes and deuterosomes), basal bodies are assembled in a process that requires CEP152 and PLK4 (although it is unclear if PLK4 activity is involved) [46,360].

Like the centriole assembly module, both PCM assembly modules (Figure 5, modules 2 and 3) may also be used independently of centrosomes as a means to tailor the MT cytoskeleton for the general and specific needs of various cell types during development and differentiation. As discussed above, the PCNT-CEP215 module and the paralogs or isoforms of both proteins are used for reassignment of the MTOC function from centrosomes to various organelles in differentiated cells [33,34,35,89,167,256,262]. Studies suggest that the PCNT-CEP192 module, by contrast, is used for the formation of mitotic acentriolar MTOCs (aMTOCs) in mouse oocytes and early embryos, which naturally lack centrosomes. In the mouse embryos, aMTOCs substitute centrosomes in organizing the mitotic spindle until the blastocyst stage, when centrioles and centrosomes are formed de novo [365]. Accordingly, aMTOCs undergo remodeling in G2-M phase through a PLK1-dependent mechanism analogous to that used for centrosome separation [50,126,366,367,368]. The hallmark of aMTOCs is the presence of CEP192, which co-localizes with AurA and PLK1 at these structures and is required for their assembly [50,369,370]. Furthermore, aMTOCs also contain other PCM proteins, such as γ-TuRC, PCNT, CEP152, and PLK4 [50,369,371,372,373,374]. Recent studies revealed that the cancerous inhibitor of PP2A (CIP2A) works together with CEP192 in promoting the assembly of aMTOC. CIP2A binds CEP192 and facilitates the recruitment of CEP192 complexes and local AurA T-loop phosphorylation at aMTOCs [370]. The presence in aMTOCs of CEP192 and its partner proteins implies that these structures comprise bona fide PCM lacking centrioles. By contrast, CEP192 does not localize to non-centrosomal MTOCs formed in differentiated cells [167,168]. The localization of CEP152 and PLK4 to aMTOCs suggests that these structures organize the centriole assembly module (Figure 5, module 1), but this module is rendered inactive—through an unknown mechanism—until after the blastocyst stage [374]. In addition, the CEP152-PLK4 complex, independently of its role in centriole assembly, facilitates MT-nucleating activity of aMTOCs, possibly by facilitating the recruitment of other PCM proteins [374,375].

Experimental inactivation of either the centriole assembly module (through inhibition of PLK4 or removal of core centriolar proteins) or the PCM assembly module (through ablation of PLK1 or of several PCM proteins at once) in proliferating somatic cells results in centrosome loss and cell cycle arrest [93,94,259,260,261]. The fact that centrosomes are not restored after the ablation procedure indicates that the mechanisms of de novo assembly of centrioles and the PCM are non-functional in proliferating somatic cells. Hence, the canonical centrosome cycle is dominant and suppresses the assembly of centrioles and the PCM de novo. Such suppression may be required to ensure that each cell has precisely two centrosomes, which form only one interphase MTOC and two mitotic MTOCs.

Together, these observations reveal that the canonical centrosome cycle has evolved to be both modular and hierarchical: it incorporates the basic mechanisms of centriole assembly and MT nucleation under the control of the more recently evolved PCM assembly mechanisms and proteins. CEP192 appears to be on top of this hierarchy, as evidenced by both phylogenetic and experimental evidence (reviewed in [40,200,376]). The unique scaffolding properties of CEP192 make this protein a central hub in the centrosomal regulatory network and a key integrator of the centriole assembly module and the PCM assembly module (Figure 5). Conceivably, the Cep192 complex has evolved by analogy with its phylogenetically older counterpart, the chromosomal passenger complex [58,200,377]. The two complexes act as analogous signaling hubs at distinct mitotic signaling platforms operating at the minus ends and plus ends of spindle MTs, respectively, consistent with the hypothesis that centrosomes and kinetochores originate from a common ancestral MTOC [42,200] (Figure 1A). Thus, the animal centrosome appears to have evolved in conjunction with the evolution of kinetochores and with the increasing reliance of cell division on the Aurora- and PLK1-signaling networks. Like the centrosome, the kinetochore is formed on principles of modularity and hierarchy, with the chromosomal passenger complex occupying the hierarchical top of the kinetochore assembly network [58,377,378,379]. Modularity and hierarchy are ubiquitous, organizing principles in biology and the main drivers of the evolution of complex organisms [380,381,382,383]. Modular and hierarchically wired networks evolve as a result of a selective pressure to reduce the number of connections, which come at a price for biological systems (as connections have to be established and maintained). Because modular and hierarchically wired networks have fewer connections, they have higher overall performance, adaptability, and evolvability [382,383]. Such properties of the centrosome biogenesis networks are consistent with centrosome variability between different lineages and with the diversity of functional states of the centrosome within the same organism.

## 11. Conclusions and Outlook

Research on centrosomes and primary cilia has been gaining momentum in the last two decades. Here, we have summarized current advances in this area, highlighting the evidence that the juxtanuclear MTOC (what is usually called the centrosome) and the primary cilium may represent two different, non-mutually exclusive, architectures of the same hybrid organelle, the animal centrosome. Indeed, mounting evidence suggests that primary cilia can only be formed by centrosomes, whereas the motile ciliogenesis requires basal bodies (which can be generated de novo), but not centrosomes [46,348,349,350,360]. Moreover, it appears that the primary ciliogenesis—which was thought to be restricted to quiescent or differentiated cells—is a part of the canonical centrosome cycle in most proliferating cells [205,211]. On the other hand, in various postmitotic differentiated cells, such as those of the sensory organs, the migrating neurons and fibroblasts, and cells of the immune system, centrosomes associate with the plasma membrane and organize primary cilia or their equivalents, immune synapses, while, at the same time, acting as MTOCs [15,221,235,247,254].

Together, these findings imply that the merger of the basal body/cilium module and the PCM/centrosome module was a major evolutionary innovation in the Amorphea, which endowed the composite centrosome with the ability to carry out specific and distinct functions at two different compartments–the juxtanuclear space and the plasma membrane. This innovation may have provided a new way of integrating the extracellular and intracellular signals and the cytoskeleton. Conceivably, the animal centrosome (and, possibly, the canonical centrosomes in other eukaryotic lineages) evolved through convergent evolution from two ancestral, physically connected MTOCs under the selective pressure to have a single MTOC capable of alternating between the nuclear membrane and the plasma membrane (Figure 1G). Oscillations between the two functional states of the animal centrosome may have an important role in the establishment of cell polarity and in tissue morphogenesis and organogenesis during development and regeneration.

Thus far, centrosomes and primary cilia have been mostly studied as separate organelles. As follows from the above analysis, further progress will require the application of a holistic approach aimed at understanding how the two structural-functional modules of the animal centrosome work together and with the cell cycle machinery during cell proliferation, migration, and differentiation. The numerous questions, which arise from the current studies, open up at least three lines for future research. First, a major effort should be devoted to exploring the centrosome/primary cilium interface in order to understand how the decisions on whether to form a juxtanuclear MTOC or a primary cilium are made and how the PCM contributes to primary ciliogenesis and ciliary signal transduction. This line of research also involves investigating the mechanisms underlying the centrosome- and primary cilium-dependent G1-S checkpoint and its loss (or attenuation) in cancer cells and in cells in which the primary cilium persists through mitosis [17,207,216,218,219,220,384]. Second, it is essential to decipher how the MTOC function (and other functions) of centrosomes is/are regulated in different contexts. The hypothetical dual-circuit model of centrosome-driven MT assembly (Figure 8) should be tested experimentally. In this regard, it will be crucial to elucidate the mechanisms underlying the attenuation of the CDK1^Cyclin B^-APC/C^Cdc20^ oscillator and of the centrosomal MTOC activity in quiescent and differentiated cells, as well as the mechanisms that enable the centrosome to simultaneously organize a primary cilium and an astral MT array. Understanding the role of CDK1 and PLK1 in the maintenance of centrosomes and primary cilia and in postmitotic differentiation programs is important not only for basic research but also for clinical practice because both these kinases are considered as attractive targets for cancer therapy [385,386,387,388]. Indiscriminate inhibition of CDK1 and PLK1 activity may interfere with the function of differentiated cell populations, leading to side effects. Third, it will be important to unravel the mechanisms that enable the distinct modules of the centrosome assembly program (Figure 5) to either operate independently of each other and of centrosomes in certain cell types (such as multiciliated and other postmitotic cells, spermatozoa, oocytes, and early embryos) or to be organized in a hierarchical order in the canonical centrosome cycle (in most somatic cells). The mouse oocytes and early embryos appear to be a valuable system for addressing these questions because it allows investigation of the three naturally occurring processes: the centrosome inactivation in oocytes, the formation of aMTOCs in oocytes and early embryos, and the PCM-mediated de novo centriole assembly in the blastocyst stage embryos [365].

Understanding the centrosome-primary cilium interface has broad implications for clinical practice, in particular for oncology, hereditary disorders, and regenerative medicine. The supernumerary centrosomes and/or the upregulation of centrosome function—often associated with the concurrent loss of primary cilia—is a hallmark of most cancers [17,21,22,233,234]. In light of the above analysis, this cancer trait can be the manifestation of a shift to a cell-autonomous signaling mode. Pharmacological manipulation of the centrosome function (if feasible) may be exploited for the development of novel anticancer therapies. One possible approach is to inhibit the proliferation of cancer cells by restoring/reinforcing the checkpoints that respond to centrosomal abnormalities and/or abnormal mitoses [259,260,261,384]. Another strategy may involve the pharmacological restoration of primary ciliogenesis in cancer cells (if necessary/possible) in conjunction with the specific targeting of the ciliary receptors and signaling pathways involved in tumor growth.

Consistent with the essential role of centrosomes in cell proliferation, polarity, and migration, these organelles are crucial for tissue regeneration in animals. Remarkably, the planarians—which are the only known animals that lack centrosomes—can regenerate a small fragment of any part of the body into a new worm [46,389,390]. Moreover, a single planarian pluripotent cell is capable of rescuing a lethally irradiated worm [390,391]. Such an extraordinary regenerative ability of planarians was suggested to be due to the loss of centrosomes and, hence, loss of the dependence of tissue remodeling on these organelles [68]. In vertebrates, the ability to regenerate heart tissue appears to correlate with the presence of centrosomes. The loss of cardiac regeneration in mammals after birth has been linked to the loss of centrosome integrity (and of primary cilia) [262]. Conceivably, the centrosome inactivation in postnatal cardiomyocytes triggers a p53-dependent G1 cell-cycle arrest that renders cells postmitotic [262,384,392]. Thus, success in the induction of proliferation of resident postnatal cardiomyocytes—which is a promising approach to heart regeneration [393,394,395]—appears to critically depend on the restoration of functional centrosomes and/or abrogation of the centrosome-dependent cell-cycle arrest without compromising the fidelity of spindle assembly and mitosis.

In conclusion, the importance of integrative research on centrosomes and primary cilia for biology and medicine can hardly be overestimated. The fact that recent studies challenge the existing paradigms may be an indication that we are on the cusp of major breakthroughs in our understanding of the biogenesis and role of both structures. 

## Figures and Tables

**Figure 1 cells-08-00701-f001:**
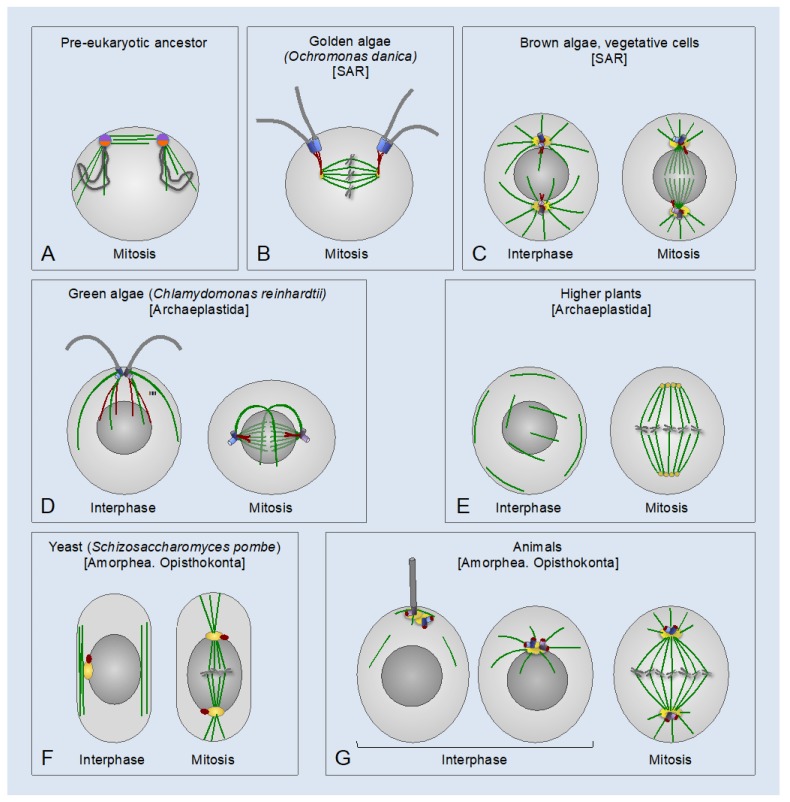
Centrosomes and the basal body apparatus in different eukaryotic lineages. (**A**) Putative pre-eukaryotic ancestor, which had circular chromosomes (dark grey loops) associated with a precursor centrosome with a dual centrosome and kinetochore function (purple and orange half-circles, respectively). The precursor centrosome was still attached to the surface membrane [42]. Microtubules (MTs) are in green. (**B**–**G**) Eukaryotes of different lineages. Centrioles/basal bodies are in blue or light purple; flagella are in grey; microtubules (MTs) are in green; pericentriolar material (PCM) is in yellow; the centrin-containing structures are in red. Higher plants (**E**) lack Polo-like kinase 1 (PLK1) and the apparent orthologs of the PCM proteins involved in γ-tubulin ring complex (γ-TuRC) anchoring and activation in animals. Conceivably, plant-specific γ-TuRC-anchoring and activating factors form centrosome-like MT-organizing centers (MTOCs), which organize spindle poles in higher plants [(yellow circles in (**E**)] [66]. Taxonomic supergroups are indicated in square brackets. SAR: stramenopiles, alveolates, and Rhizaria.

**Figure 2 cells-08-00701-f002:**
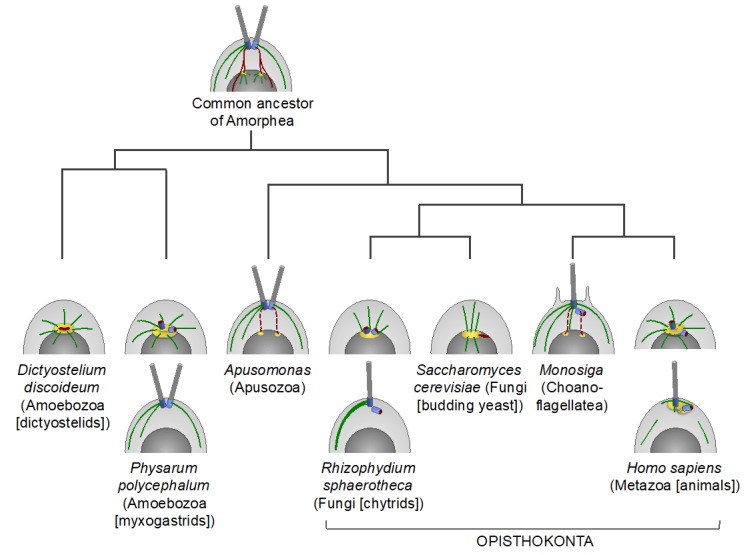
Centrosomes and the basal body apparatus in certain lineages of the Amorphea supergroup. A schematic illustration of cells in early interphase. Centrioles and basal bodies are in blue, flagella are in grey, microtubules (MTs) are in green, and the pericentriolar material (PCM) [which presumably originates from the ancestral nucleus-associated MT-organizing center (MTOC)] is in yellow. The ancestral centrin-containing nucleus-basal body connector and other centrin-containing structures are in red. Dashed red lines indicate that evidence of a nucleus-basal body connection is incomplete. In apusomonads, the basal bodies are connected to the nucleus with a striated fibrous root, rhizostyle, but it is unclear if it contains centrin or not [70]. In choanoflagellates, prior to cell division, the basal bodies duplicate and migrate to poles of the nucleus [71]. For *Physarum polycephalum* and *Rhizophydium sphaerotheca*, interphase cells of two different life cycle stages are shown. It is unclear if the basal bodies are surrounded by the PCM in these organisms.

**Figure 3 cells-08-00701-f003:**
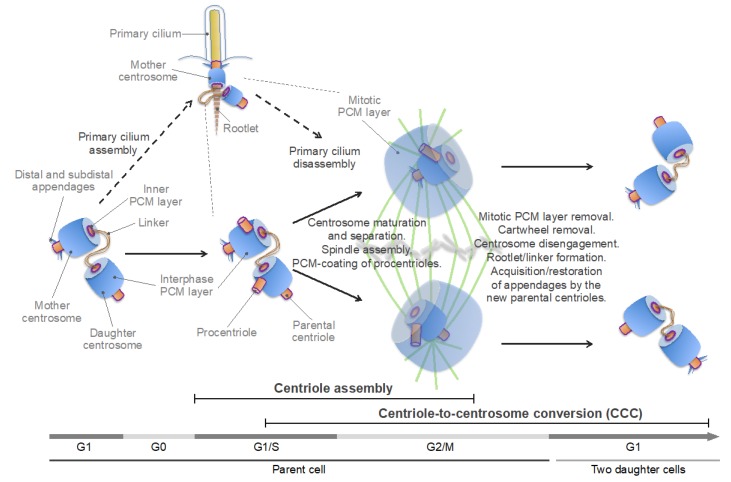
Schematic of the centrosome cycle. The inner (proximal to the centriole) pericentriolar material (PCM) layer, which contains centrosomal protein of 192 kDa (CEP192) and polo-like kinase 1 (PLK1), is highlighted by a purple line. See text for details.

**Figure 4 cells-08-00701-f004:**
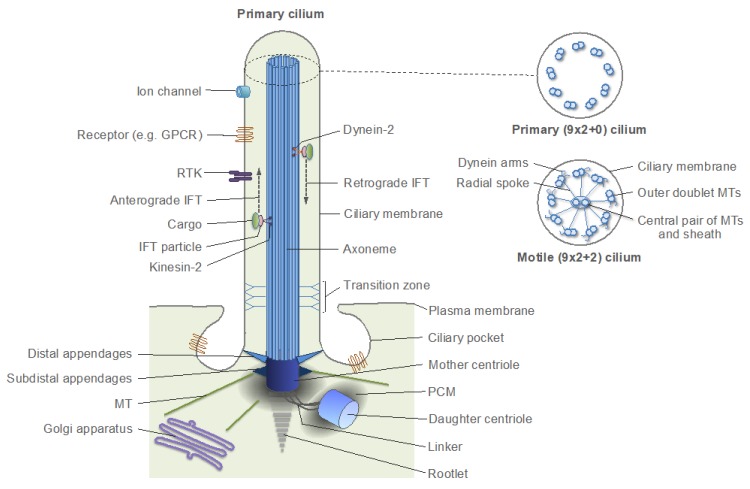
Schematic structure of a primary cilium. Panels on the right are schematic cross-sections of a typical primary cilium and a motile cilium. GPCR: G protein-coupled receptor; RTK: receptor tyrosine kinase; IFT: intraflagellar transport. MT: microtubule; PCM: pericentriolar material.

**Figure 5 cells-08-00701-f005:**
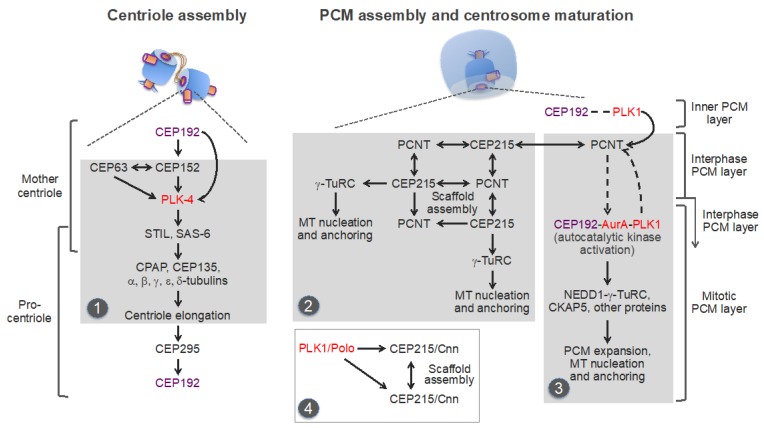
Molecular pathways underlying centriole and pericentriolar material (PCM) assembly in the vertebrates. Protein kinases are in red. The centrosomal protein of 192 kDa (CEP192) and the inner (proximal to the centriole) PCM layer, which contains CEP192 and polo-like kinase 1 (PLK1), are in purple. Distinct modules, which are repurposed and used in centrosome-independent processes, are highlighted in grey. Module 1: Basal body assembly module used in multiciliated cells. Module 2: pericentrin (PCNT)-CEP215 module used for the assembly of non-centrosomal microtubule (MT)-organizing centers (MTOCs) in postmitotic cells. Module 3: A putative PCNT-CEP192 module used for the assembly of acentriolar mitotic MTOCs in mouse oocytes and early embryos. This module relies on the CEP192-mediated, autocatalytic mechanism of Aurora A (AurA)-PLK1 activation, and PCM protein recruitment. Dashed arrows/lines indicate inferred interactions/effects that need to be experimentally validated. Module 4: PCM scaffold assembly module used in *D. melanogaster* cells. STIL: SCL-interrupting locus protein [anastral spindle 2 (Ana-2) in *D. melanogaster*; spindle assembly abnormal protein 5 (SAS-5) in *C. elegans*]; γ-TuRC: γ-tubulin ring complex; CPAP: centrosomal P4.1-associated protein [also known as centromere protein J (CENPJ); SAS-4 in *D. melanogaster* and *C. elegans*]; NEDD1: developmentally down-regulated protein 1; CKAP5: cytoskeleton-associated protein 5 [also known as colonic and hepatic tumor overexpressed protein (chTOG) and *Xenopus* MT-associated protein of 215 kDa (XMAP215)]. See text for details.

**Figure 6 cells-08-00701-f006:**
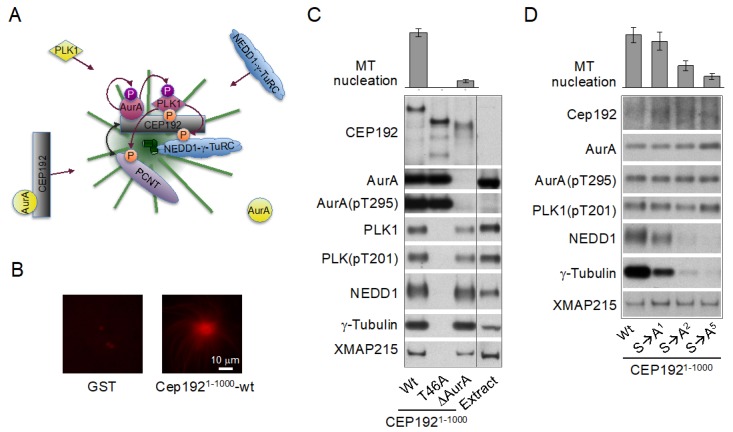
The centrosomal protein of 192 kDa (CEP192) organizes Aurora A (AurA) and Polo-like kinase 1 (PLK1) in a kinase cascade that drives microtubule (MT)-organizing center (MTOC) formation. (**A**) Schematic of the cascade. (**B**) Artificial centrosomes formed by magnetic beads coated with a recombinant N-terminal fragment of CEP192 (amino acids 1–1000) wild type (wt) (CEP192^1-1000^-wt), which binds AurA, PLK1, NEDD1-γ-TuRC (developmentally down-regulated protein 1-γ-tubulin ring complex), and *Xenopus* MT-associated protein of 215 kDa (XMAP215) [also known as cytoskeleton-associated protein 5 (CKAP5)] in a metaphase-arrested *Xenopus* egg extract. Beads coated with glutathione S-transferase (GST) are shown as a control. The extract was supplemented with rhodamine-labeled α/β-tubulin to visualize MTs. (**C**) Western blots of proteins retrieved from a metaphase-arrested *Xenopus* egg extract with beads coated with CEP192^1-1000^-wt or with its mutant counterparts lacking the binding sites for PLK1 (T46A) or AurA (δAurA). (**D**) Western blots of proteins retrieved from a metaphase-arrested *Xenopus* egg extract with beads coated with CEP192^1-1000^-wt or with its mutant counterparts lacking one (S→A^1^), two (S→A^2^), or five (S→A^5^) NEDD1-γ-TuRC-binding serines. AurA(pT295) and PLK1(pT201): AurA and PLK1 isoforms phosphorylated at the conserved threonine residue in the T loop. The graphs in (**C)** and (**D**) show a relative efficiency of MT nucleation (proportion of bead-induced MT asters). Extracts analyzed by Western blotting in (**C**,**D**) were supplemented with nocodazole to prevent MT assembly. All images are adapted from [134].

**Figure 7 cells-08-00701-f007:**
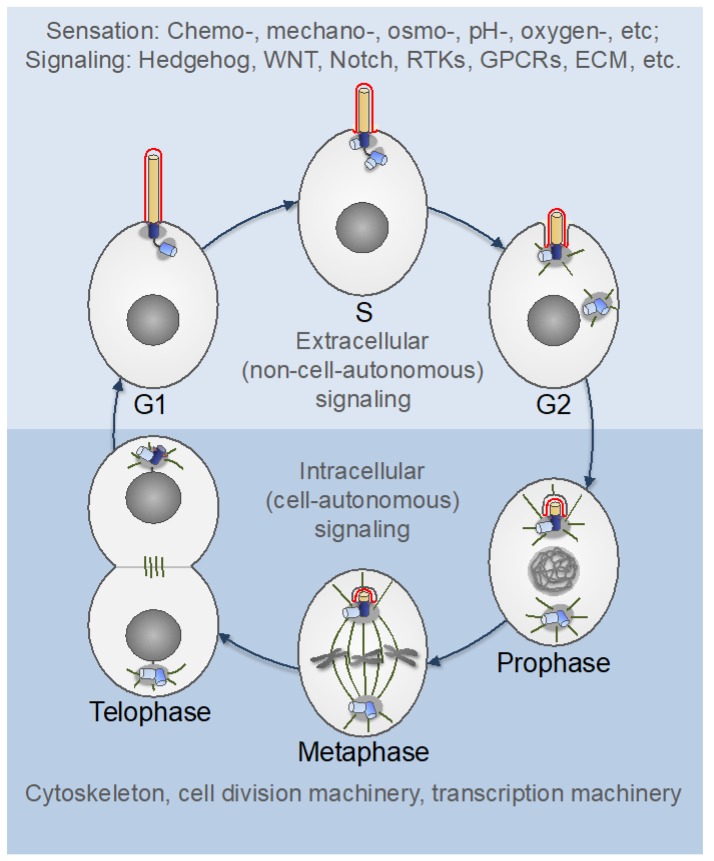
The centrosome cycle in proliferating cells. The primary cilium is formed by the mother centriole-centrosome complex in G1 phase and progressively shortens thereafter. In the interphase (upper part), the primary cilium serves as an “antenna” that senses extracellular cues and relays the signals to the cell’s interior. The ciliary membrane (red) differs in its composition from the plasma membrane and is enriched in specific ion channels and receptors for various extracellular regulatory factors (see Figure 4). After mitotic commitment (G2), the two centrosomes separate, recruit additional PCM components and form microtubule (MT) asters—the nascent spindle poles (centrosome maturation). The mother centrosome internalizes with the primary cilium while retaining the ciliary membrane, which may act as a cell fate determinant. The ciliary disassembly completes at the end of mitosis, although the timing may differ between cell types [208,211,214,216]. WNT: wingless-type MMTV integration site family; RTKs: receptor tyrosine kinases; GPCRs: G protein-coupled receptors; ECM: extracellular matrix.

**Figure 8 cells-08-00701-f008:**
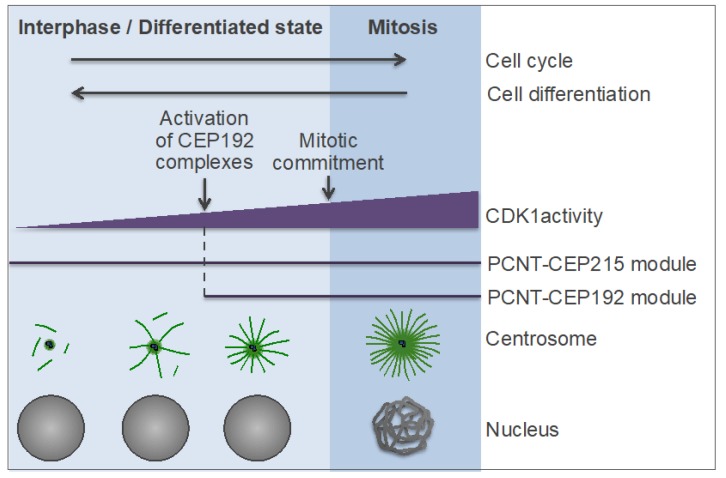
Hypothetical model of centrosome-driven microtubule (MT) assembly in interphase and postmitotic cells. See also Figure 5. CDK1: cyclin-dependent kinase 1; PCNT: pericentrin; CEP: centrosomal protein.

**Table 1 cells-08-00701-t001:** Duality of the animal centrosome.

Characteristics of the Structure	Structure Organized by the Animal Centrosome	
	Juxtanuclear MTOC (Centrosome)	Primary Cilium
Position in cells	Juxtanuclear/Spindle poles	Plasma membrane
Type	Non-membrane-bound	Membrane-bound
Principal functional component	PCM	Centrioles
Cell cycle phases in which the structure is generated	S-G2-M	G1-S-G2 (maturation in M)
Cell cycle phases in which the structure is functional	Mitosis and interphase (sometimes quiescence)	Interphase and quiescence
Putative ancestral structure	Nuclear membrane-associated MTOC for spindle assembly	Plasma membrane-associated MTOC for motile cilium/flagellum
Basic function attained from the ancestral structure	Cell division (spindle assembly), cytoskeleton organization	Sensation, intercellular communication
Signaling mode	Cell-autonomous	Non-cell-autonomous
Role in the functional organization of tissues and organs	Cell polarity and individuation	Cellular diversity and spatial organization

MTOC: microtubule (MT)-organizing center; PCM: pericentriolar material.

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
