# Peer review of "The Centrosome and the Primary Cilium: The Yin and Yang of a Hybrid Organelle"

_cells, 2019, doi:10.3390/cells8070701_

Round 1
Reviewer 1 Report
This review covers the dual roles of the centrosome complex in its functions as both a centrosome/MTOC and as a cilium that is templated by the basal body. The section on the evolution of the centrosome, basal body, and their relatedness is especially clear and instructive. The authors promote the idea that the basal body/cilium and the centrosome merged in the early evolution of Amorphea, becoming a keystone event in animal evolution.
Overall the review is very well-written and organized and covers the field deeply but concisely. I enjoyed reading it and learned a lot too. I would highly recommend this review to those in the field and especially to those beginning to do research on centrosomes and cilia.
I have only minor suggestions on revising the manuscript:
To the editors: Throughout the text, Greek letters (symbols, like alpha, beta and gamma (α, β, γ), and also dashes) appear as a strange character (). This likely happened during formatting of the manuscript.
To the authors:
In referring to reviews that describe non-centrosomal MTOCs that utilize γ-tubulin (or not) (line 63), some additional and more recent citations are worth mentioning ( doi: 10.1242/dev.153171, DOI: 10.1016/j.ceb.2016.09.003and DOI: 10.3390/cells7090121).
On line 67, I am unclear about this statement: “Hence, the basal body complex/apparatus of protists is sometimes considered both as a MTOC and a centrosome (27-29).”. An MTOC is not always a centrosome, as described in the manuscript, but when is a centrosome not an MTOC? Maybe more appropriate to say that it is a centrosomal MTOC?
In the paragraph that begins on Line 70, where the centrosome is defined is it important to restrict the definition of a centrosome to being in the cell center or being associated with the nucleus in interphase? There are examples of centrosomes that do neither (eg T lymphocyte or Drosophilaneuroblast), and the sentence following these definitions immediately introduces exceptions (“As we shall see below, the animal centrosome is unique in that it associates with either the nuclear membrane or the plasma membrane.”). Moreover, there are situations where centrosomes are inactivated, but are otherwise structurally a centrosome that has lost its MTOC activity (eg starfish oocyte, C. elegans epidermal cells, and Drosophila trachea). This is further complicated by the diversity of centrosome structures across species. Therefore, it is probably important here to be very flexible in making definitions. I know the authors are aware of these “semantic ambiquities”, so maybe in this paragraph it should be further emphasized that there are exceptions for each definition.
Figure 1: in panel A and associated legend, the dark grey loops appear to be circular chromosomes but are not indicated in the legend. Since this representation of the primordial ancestor is key to the idea being expressed, this panel should be more clearly described. In panel C, I think the spindle MTs are missing or hidden behind the nucleus layer. In panel D, there is a box with “??” in the depiction of the interphase cell (might have been a formatting issue). The legend states that centrioles and basal bodies are colored blue, but they are not.
Figure 2: The legend states that centrioles and basal bodies are colored blue, but they are not. The word “OPISTHOKONTA” has a box with a “?” inside it (likely formatting issue).
The tether that connects the mother and daughter centrioles is described briefly in the text as a Rootletin plus C-Nap1 complex at the bottom of p. 6 and illustrated in Figure 3, and maybe could be elaborated on more (in the text). Moreover, in the section on the evolution of the centrosome, the authors refer to the rhizoplast as a connector between the nucleus and the basal bodies. It seems that such “rootlets”, which seem very divergent in their protein compositions (centrins in algae, rootletin/CNap-1 in vertebrates, striated fiber assemblins (SFAs) in Apicomplexa) are a key factor that might deserve some discussion in the review including the molecular connections between these and the interconnecting fibers (‘tethers’) illustrated in Figure 3. Maybe add something about centriole cohesion and how the linker components like rootletin and C-Nap1 switch between assembling interconnecting fibers between mother centrioles and organizing ciliary rootlets (if the authors agree that it will enhance the review). Rootlets are highly variable among cell types and their functions are unclear in animals; likewise, there is no clear indication whether cohesion is important or necessary.
Since Drosophilaand C. elegansorthologs are included throughout, should add that Ana1 is the Drosophila CEP295 ortholog (Line 288). See 10.1038/ncb3274.
Reword this sentence on Line 366: “Experimental studies suggest that the recruitment is mediated by the PLK1-phosphorylated PCNT (although this needs to be experimentally confirmed) (103, 162, 163).” Sounds awkward to say that ‘experimental studies need to be experimentally confirmed’.
Figure 6 has font problems in the figure and the legend, again involving the Greek characters (they appear as a,b and g instead of α, β and γ).
The sentence ending on Line 599 should also cite the original findings on patronin (doi: 10.1016/j.cell.2010.09.022.).
Line 657: “The centrosomes are dominant MTOCs not only with regards to the extent of their own MT-nucleating capacity, but also in the sense that the formation of centrosomal MTOCs suppresses MT nucleation at all other non-centrosomal sites (135, 160, 161).” Maybe adjust the wording on this statement, as some might interpret that non-centrosomal MTOCs are inhibited by functional centrosomes. The golgi and mitochondrial MTOCs can co-exist with the centrosome, and there is evidence that they compete with each other, as indicated in the cited papers. Also see doi: 10.7554/eLife.33864.
Table 1 is a little confusing at first. Perhaps the left column, which is a set of categories that describe the second and third columns, could be distinguished by another color or by a double line or some feature to distinguish it.
Misspelled words and grammar:
“Drosophila” needs to be italicized is several places.
Line 16: add comma after “evolved”
Line 202: remove both commas: “…a process, during which centrosomes duplicate, while…”
Line 215: add “the” before “cytoplasm”.
Line 233: add “the” before “MT aster”.
Line 397: “complex” (sp).
Line 503: “juxtanuclear” (sp).
Line 640: remove comma after “concentration”.
Line 653: remove comma after “photoreceptors)”.
Line 806: “multiciliated” (sp).
Line 922: remove comma after “deuterosome-independent”.
Line 954: “centrioles” (sp).
Line 966: “phylogenetically” (sp).
Author Response
Response to Reviewer 1 Comments
This review covers the dual roles of the centrosome complex in its functions as both a centrosome/MTOC and as a cilium that is templated by the basal body. The section on the evolution of the centrosome, basal body, and their relatedness is especially clear and instructive. The authors promote the idea that the basal body/cilium and the centrosome merged in the early evolution of Amorphea, becoming a keystone event in animal evolution.
Overall the review is very well-written and organized and covers the field deeply but concisely. I enjoyed reading it and learned a lot too. I would highly recommend this review to those in the field and especially to those beginning to do research on centrosomes and cilia.
We thank the Reviewer for his/her positive evaluation and comments on the manuscript and for the constructive remarks and suggestions, which helped us to improve the paper.
I have only minor suggestions on revising the manuscript:
To the editors: Throughout the text, Greek letters (symbols, like alpha, beta and gamma (α, β, γ), and also dashes) appear as a strange character (). This likely happened during formatting of the manuscript.
à(1.1) We have noticed problems with Greek letters throughout the text, which may have occurred during reformatting of the manuscript. These problems are likely because in the original manuscript, we used a Symbol font—and not Unicode—to type Greek letters. The necessary corrections have been made throughout the text of the revised manuscript. It is desirable, if possible, not to reformat the revised manuscript.
To the authors:
In referring to reviews that describe non-centrosomal MTOCs that utilize γ-tubulin (or not) (line 63), some additional and more recent citations are worth mentioning ( doi: 10.1242/dev.153171, DOI: 10.1016/j.ceb.2016.09.003and DOI: 10.3390/cells7090121).
à(1.2) The additional references suggested by the Reviewer are now mentioned the aforementioned paragraph of the manuscript.
On line 67, I am unclear about this statement: “Hence, the basal body complex/apparatus of protists is sometimes considered both as a MTOC and a centrosome (27-29).”. An MTOC is not always a centrosome, as described in the manuscript, but when is a centrosome not an MTOC? Maybe more appropriate to say that it is a centrosomal MTOC?
à(1.3) The sentence has been changed, per Reviewer’s suggestion, as: “Hence, the basal body complex/apparatus of protists is sometimes considered as a centrosomal MTOC …”.
In the paragraph that begins on Line 70, where the centrosome is defined is it important to restrict the definition of a centrosome to being in the cell center or being associated with the nucleus in interphase? There are examples of centrosomes that do neither (eg T lymphocyte or Drosophila neuroblast), and the sentence following these definitions immediately introduces exceptions (“As we shall see below, the animal centrosome is unique in that it associates with either the nuclear membrane or the plasma membrane.”). Moreover, there are situations where centrosomes are inactivated, but are otherwise structurally a centrosome that has lost its MTOC activity (eg starfish oocyte, C. elegans epidermal cells, and Drosophila trachea). This is further complicated by the diversity of centrosome structures across species. Therefore, it is probably important here to be very flexible in making definitions. I know the authors are aware of these “semantic ambiquities”, so maybe in this paragraph it should be further emphasized that there are exceptions for each definition.
à(1.4) In order not to restrict the definition of a centrosome to the two criteria [(i) and (ii)] mentioned by the Reviewer, the word “ability” was originally used: [“ability to form a MTOC at the cell center…” and ”ability to associate with the nucleus in interphase…”], which we thought would let the reader discern that a centrosome inherently possesses the ability to localize to the cell center and associate with the nucleus, but does not always manifest these properties. Given the Reviewer’s concerns, we have modified the text accordingly: “As we shall see below, the properties (i) and (ii) of the animal centrosome (i.e. the localization to the cell center and the association with the nucleus) may not manifest in some instances, e.g. during primary ciliogenesis or immune synapse formation.”
Figure 1: in panel A and associated legend, the dark grey loops appear to be circular chromosomes but are not indicated in the legend. Since this representation of the primordial ancestor is key to the idea being expressed, this panel should be more clearly described. In panel C, I think the spindle MTs are missing or hidden behind the nucleus layer. In panel D, there is a box with “??” in the depiction of the interphase cell (might have been a formatting issue). The legend states that centrioles and basal bodies are colored blue, but they are not.
à(1.5) The figure legend for panel (A) has been modified in accordance with the Reviewer’s suggestions and now includes also a reference to the paper, which suggests the model of DNA segregation in a pre-eukaryotic ancestor depicted in the figure. All other problems with Figure 1 mentioned by the Reviewer appear to be related to the software compatibility issue. We have noticed that the problems, as mentioned by the Reviewer, appear when the manuscript file is opened on Android cell phones (with Microsoft Word APK), but not on Mac (with Microsoft Office). When the file is converted to PDF format, there are no problems with displaying Figure 1 on either Android or Mac.
Figure 2: The legend states that centrioles and basal bodies are colored blue, but they are not. The word “OPISTHOKONTA” has a box with a “?” inside it (likely formatting issue).
à(1.6) These are also problems related to the software compatibility issue [our comment (1.5)].
The tether that connects the mother and daughter centrioles is described briefly in the text as a Rootletin plus C-Nap1 complex at the bottom of p. 6 and illustrated in Figure 3, and maybe could be elaborated on more (in the text). Moreover, in the section on the evolution of the centrosome, the authors refer to the rhizoplast as a connector between the nucleus and the basal bodies. It seems that such “rootlets”, which seem very divergent in their protein compositions (centrins in algae, rootletin/CNap-1 in vertebrates, striated fiber assemblins (SFAs) in Apicomplexa) are a key factor that might deserve some discussion in the review including the molecular connections between these and the interconnecting fibers (‘tethers’) illustrated in Figure 3. Maybe add something about centriole cohesion and how the linker components like rootletin and C-Nap1 switch between assembling interconnecting fibers between mother centrioles and organizing ciliary rootlets (if the authors agree that it will enhance the review). Rootlets are highly variable among cell types and their functions are unclear in animals; likewise, there is no clear indication whether cohesion is important or necessary.
(1.7) We thank the Reviewer for bringing up this very important and interesting aspect of centrosome evolution and biogenesis. Per Reviewer’s suggestion, we have addressed this topic in section 4 and in the newly written section 8 entitled “The centrosome-connecting system”. Please note that we have changed the term “tether” to “linker”.
Since Drosophila and C. elegans orthologs are included throughout, should add that Ana1 is the Drosophila CEP295 ortholog (Line 288). See 10.1038/ncb3274.
à(1.8) The addition suggested by the Reviewer has been made.
Reword this sentence on Line 366: “Experimental studies suggest that the recruitment is mediated by the PLK1-phosphorylated PCNT (although this needs to be experimentally confirmed) (103, 162, 163).” Sounds awkward to say that ‘experimental studies need to be experimentally confirmed’.
à(1.9) The sentence has been reworded, as suggested by the Reviewer.
Figure 6 has font problems in the figure and the legend, again involving the Greek characters (they appear as a,b and g instead of α, β and γ).
à(1.10) This is a software compatibility issue [please see our comment (1.5)].
The sentence ending on Line 599 should also cite the original findings on patronin (doi: 10.1016/j.cell.2010.09.022.).
à(1.11) The citation has been added, as suggested.
Line 657: “The centrosomes are dominant MTOCs not only with regards to the extent of their own MT-nucleating capacity, but also in the sense that the formation of centrosomal MTOCs suppresses MT nucleation at all other non-centrosomal sites (135, 160, 161).” Maybe adjust the wording on this statement, as some might interpret that non-centrosomal MTOCs are inhibited by functional centrosomes. The golgi and mitochondrial MTOCs can co-exist with the centrosome, and there is evidence that they compete with each other, as indicated in the cited papers. Also see doi: 10.7554/eLife.33864.
à(1.12) This sentence has been modified, and the paper suggested by the Reviewer has been added as a new citation.
Table 1 is a little confusing at first. Perhaps the left column, which is a set of categories that describe the second and third columns, could be distinguished by another color or by a double line or some feature to distinguish it.
à(1.13) The format of Table 1 has been modified in accordance with the Reviewer’s suggestion.
Misspelled words and grammar:
“Drosophila” needs to be italicized is several places.
Line 16: add comma after “evolved”
Line 202: remove both commas: “…a process, during which centrosomes duplicate, while…”
Line 215: add “the” before “cytoplasm”.
Line 233: add “the” before “MT aster”.
Line 397: “complex” (sp).
Line 503: “juxtanuclear” (sp).
Line 640: remove comma after “concentration”.
Line 653: remove comma after “photoreceptors)”.
Line 806: “multiciliated” (sp).
Line 922: remove comma after “deuterosome-independent”.
Line 954: “centrioles” (sp).
Line 966: “phylogenetically” (sp).
à(1.14) We thank the Reviewer for taking the time to point out grammar and spelling errors, which we have now corrected.
Reviewer 2 Report
The manuscript by Joukov and De Nicolo reviews carefully the existing literature on centrosomes and cilia. The manuscript analyzes in great depth the evolutionary origin of these structures. It is convincingly argued that the animal centrosome has evolved as a merger of a plasma-membrane-associated basal body complex and a juxtanuclear MTOC. Also, the importance of functional modules (e.g. CEP215 module and CEP192 module) among centrosome proteins are well explained. Overall, - although a bit lengthy - the manuscript should be of interest (and of encyclopedic value) for many scientists who want to familiarize with this field.
There are several points, however, that must be addressed in a revised version:
Most importantly, the authors introduce a non-standard nomenclature by using the term “centrosome” synonymously for describing “centrioles” (line 206, Figure 3, lines 240, 242, and in much of the following text throughout the manuscript). In a somewhat arrogant statement, they denounce the standard nomenclature (one centrosome containing two centrioles) as a “prevailing cliché” that is (according to them) “incorrect factually and semantically” (lines 243-245). I think that the authors have made clear how the centrosome has evolved and how it can be defined functionally, but they should not alter the nomenclature that has been accepted by the entire scientific community. The corresponding paragraph should be tuned down, and the term “centriole” should be used in a conventional manner. Otherwise, the manuscript risks to confuse scientific newcomers.
Line 308: “CCC requires passage through mitosis”. A recent publication by Kim et al (2019, J Cell Sci 132. pii:jcs225789) as well as older work by the Morrison group (Dodson et al, 2004, EMBO J 23:3864-73) has provided evidence that centrosome re-duplication may occur as early as G2/M, following cleavage of pericentrin and thereby triggering disengagement.
Line 329: gamma-TuRCs form from “five molecules of gamma-TuSCs and one molecule each of GCP4, GCP5, and GCP6”: I think this statement lacks an experimental basis; although the exact number remains to be determined, gamma-tubulin complexes may form slightly more than one full helical turn, accomodating easily more than five gamma-TuSCs (Erlemann et al, 2012, J Cell Biol 197:59-74); moreover, several publications have provided evidence for two molecules of GCP4 per gamma-TuRC (Murphy et al, 2001, Mol Biol Cell 12:3340-52; Choi et al, 2010, J Cell Biol 191:1089-95).
Line 334: “gamma-TuRCs ... mimics the plus-end of a MT...” I think this is incorrect, since recent work by McIntosh et al (2018, J Cell Biol 217:2691-2708) has provided good evidence for microtubule plus-ends displaying curved protofilaments instead of being blunt. It can simply be stated that the geometry of the gamma-TuRC surface resembles the helical geometry of 13-protofilament microtubules and may therefore act as a template.
P { margin-bottom: 0.08in; }Author Response
Response to Reviewer 2 Comments
The manuscript by Joukov and De Nicolo reviews carefully the existing literature on centrosomes and cilia. The manuscript analyzes in great depth the evolutionary origin of these structures. It is convincingly argued that the animal centrosome has evolved as a merger of a plasma-membrane-associated basal body complex and a juxtanuclear MTOC. Also, the importance of functional modules (e.g. CEP215 module and CEP192 module) among centrosome proteins are well explained. Overall, - although a bit lengthy - the manuscript should be of interest (and of encyclopedic value) for many scientists who want to familiarize with this field.
We thank the Reviewer for the positive evaluation of our work and for the insightful and constructive comments, which helped us to improve the manuscript.
There are several points, however, that must be addressed in a revised version:
Most importantly, the authors introduce a non-standard nomenclature by using the term “centrosome” synonymously for describing “centrioles” (line 206, Figure 3, lines 240, 242, and in much of the following text throughout the manuscript). In a somewhat arrogant statement, they denounce the standard nomenclature (one centrosome containing two centrioles) as a “prevailing cliché” that is (according to them) “incorrect factually and semantically” (lines 243-245). I think that the authors have made clear how the centrosome has evolved and how it can be defined functionally, but they should not alter the nomenclature that has been accepted by the entire scientific community. The corresponding paragraph should be tuned down, and the term “centriole” should be used in a conventional manner. Otherwise, the manuscript risks to confuse scientific newcomers.
à (2.1) We thank the Reviewer for raising the concern regarding the distinction between the terms “centrosome” and “centriole” and for pointing out that the sentence “It is worth noting in this regard that the prevailing cliché that a newly formed animal cell contains one centrosome with two centrioles, albeit convenient, is incorrect factually and semantically (as the term CCC implies)” might generate confusion. This sentence has been deleted.
In our manuscript, we wanted to convey the emerging view, which is supported by experimental evidence, that a newly formed animal cell contains two centrosomes, each containing a single centriole (DOI 10.1016/j.devcel.2009.07.015; doi/10.1073/pnas.1716840115). This notion is consistent with the fact that the newly-formed procentrioles undergo centriole-to-centrosome conversion, i.e. acquire the PCM and key properties of a centrosome, such as the ability to recruit gamma tubulin, nucleate MTs, and duplicate. Accordingly, the process, which occurs in G2/M and underlies bipolar spindle assembly, is called ‘centrosome separation’ (rather than e.g. ‘centrosome splitting’), implying that it involves more than one centrosome.
To prevent confusion on this issue among newcomers to the field, in the revised manuscript (section 4), we draw attention to the fact that coalescence of the two centrosomes into a single MTOC, which frequently occurs in interphase cells, is, in fact, often referred to as a single centrosome. Furthermore, in the new section of the manuscript (section 8. The centrosome-connecting system), we elaborate on the mechanisms underlying centrosome coalescence and on the significance of this process for cellular physiology.
Line 308: “CCC requires passage through mitosis”. A recent publication by Kim et al (2019, J Cell Sci 132. pii:jcs225789) as well as older work by the Morrison group (Dodson et al, 2004, EMBO J 23:3864-73) has provided evidence that centrosome re-duplication may occur as early as G2/M, following cleavage of pericentrin and thereby triggering disengagement.
à(2.2) As stated in the manuscript, the centriole-to-centrosome conversion (CCC) is a process of transformation of newly assembled centrioles into fully functional centrosomes with all properties of these organelles, one of which is the ability to duplicate. In the paper by Kim et al. mentioned by the Reviewer (doi: 10.1242/jcs.225789), centrosome reduplication was reported to occur during G2/M phase in cultured mammalian (HeLa) cells depleted of pericentrin. It appears that in this setting, a malformed, pericentrin-depleted PCM of the mother centrosome promoted the formation of multiple centrioles (Fig. 8G of Kim et al., 2019). Although these centrioles acquire some PCM components, they cannot be considered centrosomes until after mitotic exit, as the authors state in the paper: “Rather, a centriole should experience mitotic exit for conversion to a centrosome. It is possible that a novel factor is required for the induction of centriole-to-centrosome conversion. This factor may be activated only after cells exit mitosis.”
The study by Dodson et al. mentioned by the Reviewer (DOI: 10.1038/sj.emboj.7600393) reports that centrosomes can duplicate during a prolonged G2 arrest (caused by Rad51 ablation) in the absence of additional rounds of DNA replication in chicken (DT40) B cells. Centrosome reduplication during G2 phase (as reported by Dodson et al. and by Kim et al.—including the experiments with the cleavage of pericentrin) is consistent with the observations that the CDK1/mitotic cyclin levels required for centrosome duplication are lower than those required for mitosis, which allows the uncoupling of the nuclear and centrosome cycle (section 7, subsection ‘Mechanisms of centrosome inactivation during cell differentiation’).
Line 329: gamma-TuRCs form from “five molecules of gamma-TuSCs and one molecule each of GCP4, GCP5, and GCP6”: I think this statement lacks an experimental basis; although the exact number remains to be determined, gamma-tubulin complexes may form slightly more than one full helical turn, accomodating easily more than five gamma-TuSCs (Erlemann et al, 2012, J Cell Biol 197:59-74); moreover, several publications have provided evidence for two molecules of GCP4 per gamma-TuRC (Murphy et al, 2001, Mol Biol Cell 12:3340-52; Choi et al, 2010, J Cell Biol 191:1089-95).
à(2.3) The statement the Reviewer refers to has been corrected as: “…. usually consists of several laterally associated molecules of γ-TuSCs assembled together with GCP4, GCP5, and GCP6 in heterodimers with γ-tubulin (3, 4, 164)”
Line 334: “gamma-TuRCs ... mimics the plus-end of a MT...” I think this is incorrect, since recent work by McIntosh et al (2018, J Cell Biol 217:2691-2708) has provided good evidence for microtubule plus-ends displaying curved protofilaments instead of being blunt. It can simply be stated that the geometry of the gamma-TuRC surface resembles the helical geometry of 13-protofilament microtubules and may therefore act as a template.
à (2.4) An important point. The statement above has been corrected, as suggested by the Reviewer.
Reviewer 3 Report
The manuscript of Joukov and De Nicolo entitled “The Centrosome and the Primary Cilium: the Yin and Yang of a Hybrid Organelle” is a comprehensive review dealing with different functions of the centrioles during the cell life. The AA propose here an interesting hypothesis on the origin of the centriole/centrosome complex during the animal evolution and suggest to distinguish the role of the centrioles involved in centrosome/primary cilia assembly from basal bodies needed to assemble motile flagella. This difference is mainly base on the ability of the centriole to recruit PCM. Although I think that centrioles and basal bodies are different functional aspects of the same organelle, I retain that the hypothesis proposed in the present manuscript deserves attention. Moreover, in my opinion the centrioles organize the PCM and are part of the centrosome, but centrioles cannot be retained equivalent to the centrosome, unless change their definition. However, the present review is interesting and well written and deserves publication in Cells.
I have only a few minor comments:
l.82: there are also centrioles with doublets (i.e. Insects) and singlets (C.elegans).
Fig. 1E. The spindle of higher plants is acentriolar. Thus, the PCM is a large aggregate and the poles are not focused (they look like those of the early mouse embryo).
l.96: In my opinion the kinetochore cannot be retained a MTOC. The microtubules are captured by the kinetochore, but cannot be nucleated.
l. 204: better “usually synchronized”. During male gametogenesis of most animals the centrioles duplicate during prophase I in the absence of DNA replication.
l. 223: “at the proximal end of the PCM of each parental centrosome. Unclear
l. 238: why “procentriole”?
l.242: “Thus, each daughter cell inherits a pair of centrosomes, each containing a single centriole” Unclear
l.467: “implicit that the primary cilia are formed by centrosomes”. The primary cilia are assembled following the elongation of the A and B tubules of the centriole. It appears that the AA retain the centrosome and the centriole as the same structure.
l.492: “Consistent with this notion, primary cilia are not found in circulating lymphocytes and granulocytes “. All the Insects and also HeLa cells lack primary cilia.
l.628: “non-motile primary cilia with a (9 × 2 + 2)”. Primary cilia lack the central doublet as reported in Fig. 4.
Author Response
Response to Reviewer 3 Comments
The manuscript of Joukov and De Nicolo entitled “The Centrosome and the Primary Cilium: the Yin and Yang of a Hybrid Organelle” is a comprehensive review dealing with different functions of the centrioles during the cell life. The AA propose here an interesting hypothesis on the origin of the centriole/centrosome complex during the animal evolution and suggest to distinguish the role of the centrioles involved in centrosome/primary cilia assembly from basal bodies needed to assemble motile flagella. This difference is mainly base on the ability of the centriole to recruit PCM. Although I think that centrioles and basal bodies are different functional aspects of the same organelle, I retain that the hypothesis proposed in the present manuscript deserves attention. Moreover, in my opinion the centrioles organize the PCM and are part of the centrosome, but centrioles cannot be retained equivalent to the centrosome, unless change their definition. However, the present review is interesting and well written and deserves publication in Cells.
We thank the Reviewer for his/her positive evaluation and valuable comments, which helped us to improve our manuscript.
à(3.1) We believe that the presence or absence of the PCM and/or of some of its components is a critical factor that determines whether centrioles/basal bodies nucleate the formation of a primary cilium or a motile cilium/flagellum. By other words, a primary cilium can only be formed by a centriole/basal body in the context of a centrosome.
I have only a few minor comments:
l.82: there are also centrioles with doublets (i.e. Insects) and singlets (C.elegans).
à(3.2) The statement the Reviewer refers to has been corrected as: “…symmetrically arranged triplets (or, in some organisms, doublets or singlets) of stable MTs (36, 37)”.
Fig. 1E. The spindle of higher plants is acentriolar. Thus, the PCM is a large aggregate and the poles are not focused (they look like those of the early mouse embryo).
à(3.3) A great point, thanks. Figure 1E and its legend have been modified accordingly.
l.96: In my opinion the kinetochore cannot be retained a MTOC. The microtubules are captured by the kinetochore, but cannot be nucleated.
à(3.4) In our manuscript, we suggest a definition of a MTOC as any structure that generates, organizes, and/or anchors MTs. We believe that MT capture by kinetochores, in itself, can be seen as a process of organizing MT plus ends. Moreover, kinetochores not only capture MTs, but also control the dynamics of MT plus ends through the chromosomal passenger complex (CPC). In particular, Aurora B activation upon CPC clustering was shown to stabilize kinetochore MTs by suppressing MT-depolymerizing activity of MCAK (PMID: 14960279; DOI: 10.1016/j.cell.2004.06.026; DOI: 10.1016/j.cub.2004.01.055; DOI: 10.1101/gad.1945310; DOI 10.1016/j.devcel.2006.11.001; DOI 10.1016/j.cub.2009.05.061).
In the revised manuscript (section 1), we have rephrased and softened the statements regarding the role of kinetochores as MTOCs.
l. 204: better “usually synchronized”. During male gametogenesis of most animals the centrioles duplicate during prophase I in the absence of DNA replication.
à(3.5) The sentence has been corrected as suggested by the Reviewer.
l. 223: “at the proximal end of the PCM of each parental centrosome. Unclear
à(3.6) This sentence has been modified.
l. 238: why “procentriole”?
à(3.7) The word has been changed to “centriole”.
l.242: “Thus, each daughter cell inherits a pair of centrosomes, each containing a single centriole” Unclear
à(3.8) This statement has been changed to “Thus, each nascent daughter cell inherits a pair of centrosomes – each containing a single centriole – the parental one, and the one formed by the daughter centriole that has completed the CCC (Fig. 3)”. Please also see our comment (2.1) above.
l.467: “implicit that the primary cilia are formed by centrosomes”. The primary cilia are assembled following the elongation of the A and B tubules of the centriole. It appears that the AA retain the centrosome and the centriole as the same structure.
à(3.9) This statement has been changed to “it is implicit that the centriole, which assembles a primary cilium, is a part of the centrosome in proliferating cells”.
l.492: “Consistent with this notion, primary cilia are not found in circulating lymphocytes and granulocytes “. All the Insects and also HeLa cells lack primary cilia.
à(3.10) HeLa cells are cancer cells. In the revised manuscript, the following sentence has been added after the sentence mentioned by the Reviewer: “Primary cilia are also frequently absent in cancer cells, likely as a reflection of cell-signaling derangement”.
Although insects may not have primary cilia akin to those found in most somatic vertebrate cells, insects do have cilia in sensory neurons, which are considered as primary cilia (e.g. doi/10.1083/jcb.201502032, DOI 10.1186/s13630-015-0018-9, and section 8 of the revised manuscript, last sentence). The primary cilia of sensory organs may be called “specialized”, with the reservation that all primary cilia are adapted to serve each particular cell type, and, therefore, are specialized in one way or another.
l.628: “non-motile primary cilia with a (9 × 2 + 2)”. Primary cilia lack the central doublet as reported in Fig. 4.
à(3.11) Olfactory primary cilia and the kinocilium of cochlear hair cells are unique in that, unlike most primary cilia, they have a (9 × 2 + 2) MT configuration of axonemes (but they are, nevertheless, nonmotile, as they lack the dynein arms) (doi:10.3390/cells4030500; DOI 10.1007/s00018-017-2570-5). What determines the absence/presence of the central MT doublet and of the dynein arms in the cilia is an intriguing question, which remains to be addressed.